Article 

# Hepatic SREBP signaling requires SPRING to govern systemic lipid metabolism in mice and humans

Sebastian Hendrix [1], Jenina Kingma [1], Roelof Ottenhoff[1], Masoud Valiloo[1], Monika Svecla [2], Lobke F. Zijlstra [1], Vinay Sachdev[1], Kristina Kovac[1], Johannes H. M. Levels[3], Aldo Jongejan [4], Jan F. de Boer [5,6], Folkert Kuipers [5,7], Antoine Rimbert [8], Giuseppe D. Norata [2], Anke Loregger[1,9] & Noam Zelcer [1] ✉

The sterol regulatory element binding proteins (SREBPs) are transcription factors that govern cholesterol and fatty acid metabolism. We recently identified SPRING as a post-transcriptional regulator of SREBP activation. Constitutive or inducible global ablation of *Spring* in mice is not tolerated, and we therefore develop liver-specific *Spring* knockout mice (LKO). Transcriptomics and proteomics analysis reveal attenuated SREBP signaling in livers and hepatocytes of LKO mice. Total plasma cholesterol is reduced in male and female LKO mice in both the low-density lipoprotein and high-density lipoprotein fractions, while triglycerides are unaffected. Loss of *Spring* decreases hepatic cholesterol and triglyceride content due to diminished biosynthesis, which coincides with reduced very-low-density lipoprotein secretion. Accordingly, LKO mice are protected from fructose diet-induced hepatosteatosis. In humans, we find common genetic *SPRING* variants that associate with circulating high-density lipoprotein cholesterol and ApoA1 levels. This study positions SPRING as a core component of hepatic SREBP signaling and systemic lipid metabolism in mice and humans.

The liver is central to the coordination of systemic lipid metabolism, dysregulation of which is associated with development of, amongst others, non-alcoholic fatty liver disease, dyslipidemia, and cardiovascular diseases. Herein, the transcription factors sterol regulatory element binding proteins (SREBPs) play a critical role. SREBPs are produced in their precursor form as endoplasmic reticulum (ER)-resident membrane proteins. Their association in a complex with SCAP and INSIG couples their retention in the ER and subsequent proteolytic activation to the sensing of cholesterol content in the ER membrane by SCAP[1–3]. A decrease in ER-membrane cholesterol content results in the dissociation of INSIG from SCAP, and COPII-dependent anterograde transport of the SCAP/SREBP complex to the Golgi[4,5],

[1]Department of Medical Biochemistry, Amsterdam UMC, Amsterdam Cardiovascular Sciences and Gastroenterology and Metabolism, University of Amsterdam, Meibergdreef 9, 1105AZ Amsterdam, The Netherlands. [2]Department of Pharmacological and Biomolecular Sciences, Università degli Studi di Milano, Via Balzaretti 9, 20133 Milan, Italy. [3]Department of Experimental Vascular Medicine, Amsterdam UMC, University of Amsterdam, Meibergdreef 9, 1105AZ Amsterdam, The Netherlands. [4]Department of Epidemiology and Data Science, Bioinformatics Laboratory, of Academic Medical Center, University of Amsterdam, Meibergdreef 9, 1105AZ Amsterdam, The Netherlands. [5]Department of Pediatrics, University Medical Center Groningen, University of Groningen, Groningen, The Netherlands. [6]Department of Laboratory Medicine, University Medical Center Groningen, University of Groningen, Groningen, The Netherlands. [7]European Research Institute for the Biology of Ageing (ERIBA), University Medical Center Groningen, University of Groningen, Groningen, The Netherlands. [8]l'institut du thorax, Nantes Université, CNRS, INSERM, F-44000 Nantes, France. [9]Present address: Myllia Biotechnology GmbH, Am Kanal 27, 1110 Vienna, Austria. ✉e-mail: n.zelcer@amsterdamumc.nl

where SREBPs are proteolytically cleaved by the sequential activity of S1P and S2P (encoded by *MBTPS1* and *MBTPS2*, respectively)[6,7]. The released N-terminal SREBP-1 and −2 transcriptional domains translocate to the nucleus to activate transcriptional programs that promote fatty acid and cholesterol biosynthesis, respectively[8]. At the same time, SCAP and the remnant C-terminal domain of SREBP undergo retrograde transport to the ER, where remnant SREBP is degraded and SCAP can enter a new cycle of SREBP activation[1,9].

Owing to its pivotal role in control of lipid metabolism, the SREBP pathway is subject to tight regulation that integrates a wide array of metabolic inputs, signaling pathways, and (post)-transcriptional feedback mechanisms. Using a suite of genetic screens performed in mammalian haploid cells we recently identified the SREBP Regulating Gene (*SPRING/C12ORF49*) as a previously unrecognized determinant of SREBP signaling[10]. We and others have proposed that, in cultured cells, SPRING is implicated in two important regulatory steps in the SREBP pathway, i.e. the retrograde transport of SCAP and in regulating the activity of S1P[10-13]. Briefly, cells lacking SPRING have a functional depletion of SCAP in the ER due to its impaired retrograde trafficking[10]. Additionally, the proteolytic activation of S1P into its mature Golgi-associated form is attenuated[13,14]. Consequently, this limits the prerequisite first-step proteolytic activation of SREBPs[6], as well as of other S1P substrates such as ATF6, GNMTB, and CREB3L3[15-19]. While seemingly independent of each other, the two processes may actually be intertwined since pharmacological or genetic inhibition of S1P has been shown to lead to SCAP degradation[20].

Evidence that SPRING is a physiological important regulator of SREBPs is limited. We have shown that short-term adenovirus-mediated hepatic silencing of *Spring* mildly attenuates the SREBP-dependent postprandial transcriptional response[10]. Additionally, while global ablation of *Spring* and *Mbtps1* in mice is embryonically lethal[10,21], deletion of *Spring* phenocopies some features of *Mbtps1* deletion in zebrafish[12], supporting the notion that the two proteins act in a shared pathway. However, whether SPRING is a physiologically important regulator of SREBP activities in control of hepatic and systemic lipid metabolism is unknown. To address this issue we developed a liver-specific *Spring* knockout mice (LKO) model and studied its metabolic phenotype.

We show here that LKO mice display a dramatic dysregulation of hepatic SREBP signaling, and consequently disturbed lipid metabolism, thereby providing a potential explanation for the association between genetic variation in human *SPRING* and plasma lipoprotein levels reported in this study.

## Results

### Development of SPRING LKO mice

We have previously reported that in mice *Spring* is ubiquitously expressed and that constitutive global deletion of *Spring* is embryonically lethal[10]. To evaluate whether this only reflects a developmental requirement for *Spring* we attempted to globally ablate *Spring* in adult mice. For this purpose, we developed CreERT2⁻-*Spring*^(fl/fl) (control) and CreERT2⁺-*Spring*^(fl/fl) mice and induced *Spring* deletion by administering tamoxifen. While this treatment had no effect in control mice, tamoxifen-induced global deletion of *Spring* in adult CreERT2⁺-*Spring*^(fl/fl) mice resulted in lethargy, reduced body- and liver weight, eventually reaching humane endpoints that required their killing (Supplementary Fig. 1a–c). We have not attempted to identify the primary cause for the observed lethargy, but this observation suggests that complete loss of *Spring* is not tolerated. Therefore, to study the role of hepatic *Spring* in regulating the SREBP pathway and systemic lipid metabolism we generated liver-specific *Spring* knockout mice (LKO; Supplementary Fig. 1d). Ablation of hepatic *Spring* was highly effective and tissue specific (Fig. 1a), and LKO mice were born with normal mendelian distribution and showed no overt hepatic phenotype when fed normal chow (Fig. 1b–d).

### SREBP signaling is attenuated in livers of LKO mice

The transition from the fasted to the fed state is accompanied by acute reprogramming of hepatic gene expression and induction of the SREBP-regulated pathway[22]. This insulin-driven transcriptional program is required, amongst others, to prime hepatocytes to increase de novo lipogenesis (DNL) in face of enhanced substrate availability. To evaluate the role of *Spring* in this acute metabolic transition, we transcriptionally profiled livers from fasted and fasted-refed LKO and control mice (Fig. 1e and Supplementary Data 1). Gene-ontology analysis identified cholesterol biosynthesis and metabolism as the primary down-regulated pathways in refed LKO livers (Supplementary Fig. 1e), and revealed a dramatic attenuation of the complete SREBP1- and SREBP2-regulated gene programs (Fig. 1f, g). A comparable decrease in SREBP signaling was also observed in fasted mice, but as expected the overall expression of SREBP target-genes was lower than in refed animals (Supplementary Fig. 2a).

We then aimed to explore whether the changes observed in the mRNA expression of SREBP target-genes were also observed at the protein level. For this purpose, we harvested livers from control and LKO male mice and processed them for unbiased proteomics determination of protein abundance (Fig. 2a). A total of 1893 proteins were quantified, of which 253 and 187 were down- and up-regulated, respectively, in livers of LKO mice. Pathway analysis identified a reduction in the abundance of multiple proteins involved in cholesterol and fatty acid synthesis in livers of LKO mice (Fig. 2b, c and Supplementary Data 2 and 3). This analysis also identified an increase in LXR/RXR signaling, but this was based on a limited set of non-canonical LXR targets representing ~10% of the pathway. Direct immunoblotting confirmed reduced abundance of a panel of representative SREBP1 and SREBP2 targets in liver homogenates from refed male LKO mice (Fig. 2d). A similar reduction in SREBP targets was also seen in liver samples from 20 h fasted LKO mice (Supplementary Fig. 2b). Liver samples obtained from female LKO mice displayed a similar reduction in SREBP targets as seen in male LKO mice, indicating that hepatic regulation of the SREBP pathway by SPRING is sex-independent (Supplementary Fig. 3a). The observation that proteolytic maturation of SREBP is attenuated in LKO mice also provides a plausible explanation for reduced SREBP transcriptional output (Supplementary Fig. 3a), consistent with our earlier report in cell lines[10]. Intriguingly, levels of the low-density lipoprotein receptor (LDLR) were largely unchanged, despite the LDLR being a bona fide SREBP2 target and the observation that hepatic *Ldlr* expression is decreased (Fig. 1f, g). Plausibly, this reflects the steady-state balance between decreased production of LDLR and of its post-transcriptional regulator PCSK9 (Fig. 1f), as also previously reported[23,24]. Dysregulation of the SREBP pathway, due to absence of *Spring*, was not limited to the intact liver as it was recapitulated in isolated primary hepatocytes from LKO mice, both at the transcript and protein level (Fig. 2e, f). Collectively, these results indicate that SPRING is indispensable for physiologic SREBP signaling in the liver.

### Lipid metabolism is dysregulated in LKO mice

The SREBP pathway governs hepatic lipid and lipoprotein metabolism, and lesions or perturbations in this pathway have profound metabolic effects. Having established an important role for SPRING in hepatic SREBP signaling, we therefore turned to interrogate the functional consequences of hepatic *Spring* loss. The dysregulated SREBP signaling in livers of LKO mice led to marked changes in circulating lipid levels. Both male and female LKO mice showed a dramatic reduction in circulating cholesterol levels (Fig. 3a, b). This reduction was apparent in both the low- and high-density lipoprotein-associated fractions (Fig. 3c and Supplementary Fig. 3b). Enhanced hepatic lipoprotein clearance due to reduced *Pcsk9* expression may contribute to this lipid profile, yet as hepatic LDLR levels were unchanged in LKO mice it is unlikely that this is the primary underlying mechanism (Fig. 2d, f). In

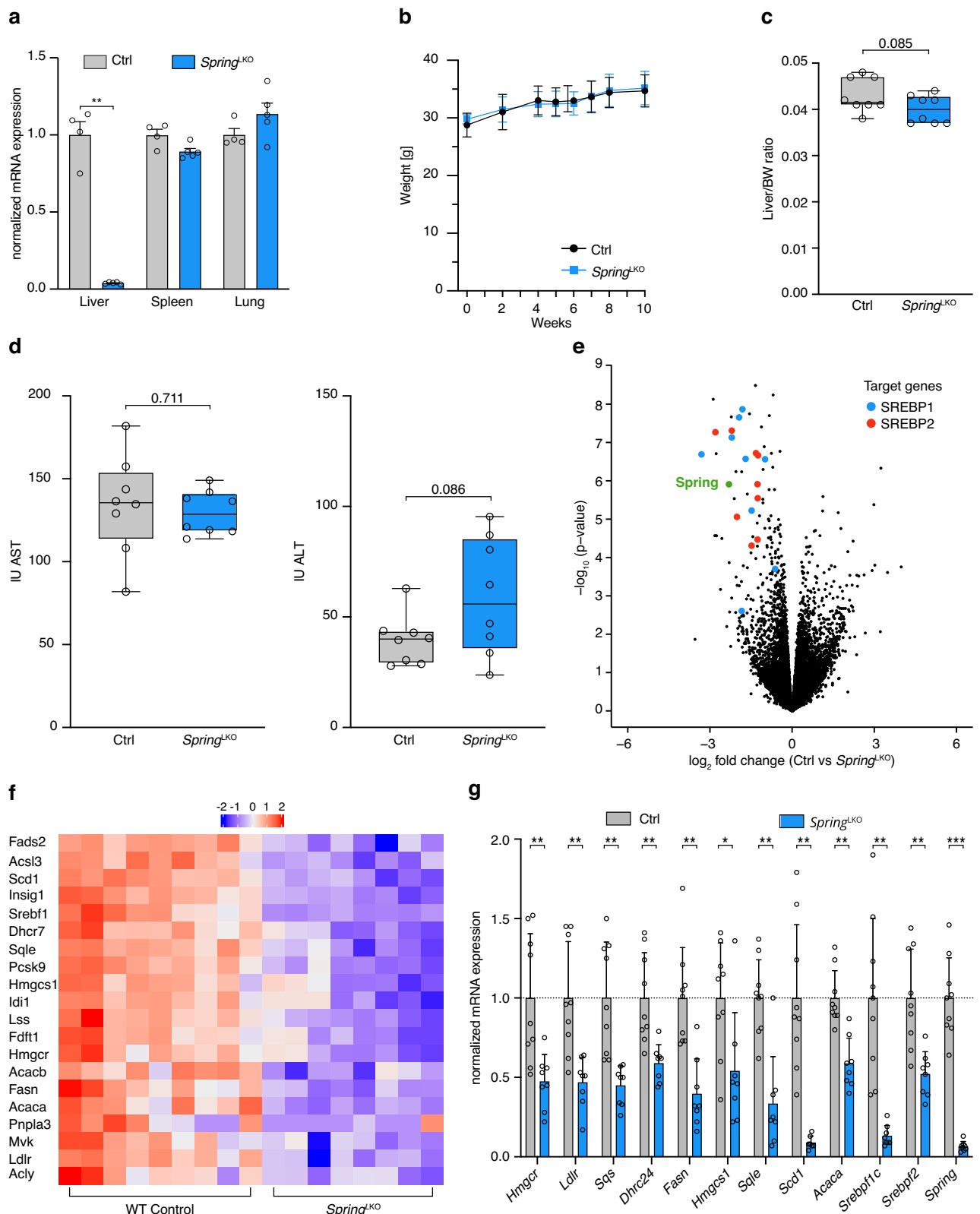

contrast to cholesterol, circulating triglyceride levels in 4 h fasted mice were comparable between LKO and control mice (Fig. 3d–f and Supplementary Fig. 3c). Unexpectedly, plasma glucose levels in 4 h fasted mice were slightly lower in LKO mice (Supplementary Fig. 4a). To explore potential explanations for this we used glucose and insulin tolerance tests to assess systemic glucose clearance and insulin sensitivity. In both assays LKO and their control counterpart mice were

comparable, indicating that a different process likely underlies the reduced glucose levels in LKO (Supplementary Fig. 4b, c).

Having established that loss of hepatic *Spring* governs SREBP signaling and systemic lipid metabolism, we aimed to address potential mechanisms linking these processes. The most conceivable explanation for this link is that attenuation of the SREBP pathway in hepatocytes would lead to decreased DNL. We first evaluated in vivo

**Fig. 1 | The hepatic SREBP-regulated postprandial program is disrupted in LKO mice. a** The indicated tissues were isolated from control (ctrl) and LKO male mice and expression of *Spring* determined by qPCR ($n = 4$ mice).p-value liver (0.0047). **b** Weight development of chow-fed 8-week-old control and LKO mice ($n = 8$ mice/group) was monitored over the indicated period. At killing the (**c**) body/liver weight ratio was determined ($n = 8$ mice/group). **d** Plasma was collected from 8-week-old control and LKO mice and the activity of AST and ALT determined enzymatically ($n = 8$). **e** Control ($n = 9$) and LKO ($n = 8$) mice were fasted for 16 hrs and subsequently refed with chow for 4 h. Transcriptional profiling of refed livers is shown as volcano plot (LKO/Control). Blue- and red-marked dots represent SREBP1 and SREBP2 targets, respectively. **f** Heat map depicting expression profiles of the indicated SREBP target genes highlighted in **e** in control and LKO refed livers. Each column represents an independent mouse. **g** Expression of the indicated genes was evaluated by qPCR in livers from control ($n = 9$ mice) and LKO mice ($n = 8$ mice). *p*-values: Hmgcr (0.009); Ldlr (0.073); Sqs (0.073); Dhrc24 (0.0073); Fasn (0.0033); Hmgcs1 (0.0192); Sqle (0.0019); Acac (0.0015); Srebpf1c (0.0055) Srebpf2 (0.0073). (**a**, **b**, **g**) Each value and error represent the mean ± SEM. (**c**, **d**) Box plots show the median (middle line), 25th, 75th percentile (box) and minimum and maximum values (whiskers). *$p < 0.05$, **$p < 0.01$ and ***$p < 0.001$ analyzed by (**a**, **b**) two-way ANOVA with Holm-Sidak post hoc analysis or (**c**, **d**) two-tailed Welch's t-test or (**g**) mixed-effects analysis with Holm-Sidak post hoc analysis. Source data are provided as a Source Data file.

hepatic fatty acid *de-novo* synthesis by providing chow-fed mice [$^{13}$C] acetate in their drinking water for 3 days ad libitum. Incorporation of heavy isotope-labeled acetate into fatty acids can be subsequently assessed by gas chromatography mass spectrometry (Fig. 4a), which then allows for the calculation of the amounts of de novo synthesized fatty acids using mass isotopomer distribution analysis. Consistent with decreased SREBP1 signaling (e.g. lower abundance of, amongst others, FASN, ACC, and SCD1 (Fig. 2b, d, f)), biosynthesis, elongation, and desaturation of fatty acids was markedly reduced as reflected by decreased levels of newly synthesized C16:0, C16:1, C18:0, and C18:1 fatty acids in LKO mice (Fig. 4b−e). To evaluate cholesterol de-novo synthesis we resorted to isolated primary hepatocytes. In LKO hepatocytes SREBP signaling is strongly diminished (Fig. 2e, f), and accordingly we measured a dramatic reduction in cholesterol synthesis to levels comparable to that obtained with inhibition of HMGCR with a statin (Fig. 4f). Lipidation of nascent ApoB-containing very-low-density lipoprotein (VLDL) particles is a key step in their production and subsequent secretion[25–27]. Our observation that DNL is reduced in LKO hepatocytes suggests that the availability of lipids for ApoB lipidation may be limited, which in turn could lead to reduced VLDL secretion and lower plasma lipid levels. To test this hypothesis, we inhibited LPL and determined the accumulation of triglycerides over time in plasma, as a proxy for hepatic VLDL production and secretion. Consistent with the proposed mechanism, the rate of plasma triglyceride accumulation in LKO mice was significantly decreased, supporting the notion that there is indeed reduced hepatic VLDL secretion in these mice (Fig. 4g, h). Importantly, impaired VLDL-triglyceride secretion did not result in increased hepatic cholesterol or triglyceride levels, as these were lower in both male and female LKO mice (Fig. 4i−l). This further supports our contention that diminished substrate availability in LKO mice limits hepatic VLDL output.

### LKO mice are protected from fructose-induced hepatosteatosis

Our findings indicate that loss of hepatic *Spring* decreases SREBP-driven signaling in chow-fed LKO mice. As a first step toward evaluating the therapeutic potential of targeting hepatic *Spring* we opted to challenge mice with a fructose-enriched diet, reasoning that this carbohydrate-rich diet stresses hepatic DNL and development of hepatosteatosis. No differences in weight gain or in the liver/body-weight ratio were observed between control and LKO mice during the dietary challenge (Fig. 5a, b). Similarly, dietary fructose did not induce changes in circulating plasma cholesterol and triglyceride levels, and these were comparable to those measured in chow-fed mice (Figs. 5c, d and c.f. 3a, d). The levels of the transaminases ALT and AST were also comparable between the control and LKO fructose-fed mice (Fig. 4e, f). Livers from control mice showed prominent punctated neutral-lipid staining, which was largely absent in livers of LKO mice (Fig. 5g, h). Intriguingly, this increased neutral lipid accumulation in control mice did not result from marked fructose-induced triglyceride accumulation (Figs. 5i and c.f. and 4j). Rather, the diet led to a dramatic increase in hepatic cholesterol(ester) content in control mice, suggesting that the high carbohydrate supply is diverted preferentially towards cholesterol biosynthesis (Figs. 5j and c.f. and 4i). In contrast to control mice, LKO mice appeared to be refractory to fructose and hepatic cholesterol content remained comparable to that in chow-fed LKO mice. In aggregate, results in LKO mice support a central role for SPRING in coordinating SREBP-driven DNL and demonstrate that the absence of *Spring* protects mice from high carbohydrate diet-induced hepatosteatosis.

### SPRING variants associated with lipid traits in humans

To extend our observations beyond mice, we explored whether genetic variation in the *SPRING (C12ORF49)* locus (12q24.22) are associated with lipid traits in humans. We first made use of data from the largest genome-wide association study (GWAS) published thus far by the Global Lipids Genetics Consortium effort, which aggregated results from ~1.6 million individuals[28]. We found genetic variants significantly associated with plasma high-density lipoprotein cholesterol (HDL-c; $p < 5.0E\text{-}08$; Fig. 6a). None of the 17 *SPRING* SNPs that significantly associated with plasma HDL-c associated with hepatic *Spring* expression, nor were they reported in the GTEx dataset as eQTLs. The top-associated single nucleotide polymorphism (SNP), rs10507274, more specifically the "C" allele, found in ~6% of the general population, is significantly associated with increased plasma levels of HDL-c ($p = 8.6E\text{-}11$, Beta = +0.0183 ( ± 2.8E-03); Fig. 6a, b). The rs10507274-C variant induces the change of a Glutamine into an Arginine at the position 55 of the SPRING protein (NM_024738.4, NP_079014.1, p.Gln55Arg), and at present the functional consequence of this coding variation is unknown. Of note, rs10507274 was associated with a reduction in plasma low-density lipoprotein cholesterol and triglyceride levels, but this did not reach genome-wide statistical significance (Fig. 6b). In order to test the association of rs10507274 with a broader panel of circulating biomarkers, we used GWAS data from ~500,000 participants from the UK Biobank. We similarly found that rs10507274-C carriers present with significantly higher plasma HDL-c and Apolipoprotein-A1 levels than non-carriers ($p = 5.3E\text{-}09$ and $p = 8.3E\text{-}16$, respectively; Fig. 6c). We did not identify any genome-wide significant association of rs10507274 with other circulating biomarkers among the 30 biomarkers tested (Supplementary Data 4). In summary, our study positions SPRING as a determinant of lipid and lipoprotein metabolism in mice and humans owing to its indispensable role in SREBP signaling.

## Discussion

Our study addresses the physiological role of the hepatic SREBP-SPRING nexus. The most important finding of our study is that hepatic *Spring* is indispensable for intact SREBP signaling, and that as a consequence, its absence leads to marked alterations in hepatic and systemic lipid metabolism. These changes provide a plausible rational to explain the association between genetic variation in *SPRING* and lipoprotein levels in humans, positioning SPRING as an important physiological determinant of hepatic and systemic lipid metabolism in mice and humans.

The results of this study further cement SPRING as a core component of the SREBP-activating machinery[1]. In mice, global deletion of *Srebf1* (encoding SREBP1) results in 50−85% embryonic

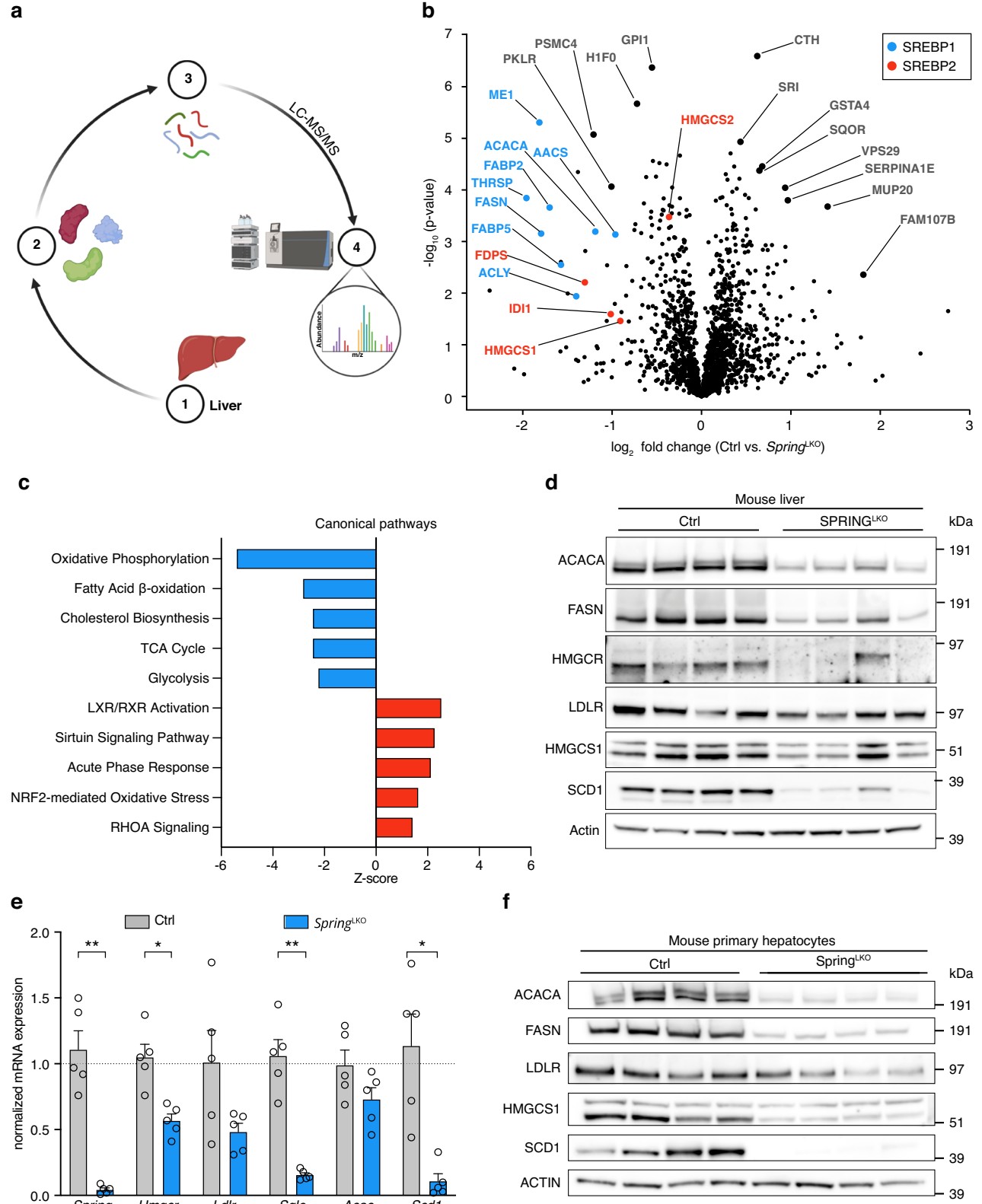

**Fig. 2 | Attenuated SREBP signaling in LKO livers and primary hepatocytes. a** Scheme depicting proteomic strategy used to identify and label-free quantify (LFQ) differentially abundant proteins in LKO mice. **b** Livers were collected from control and LKO male mice (*n* = 5/group) and subjected to unbiased proteomics analysis (4 h fasted). Volcano plot depicting significantly modulated proteins in control vs. LKO liver with indicated SREBP1 (blue) and SREBP2 (red) regulated targets highlighted. **c** Pathway analysis shows the top 5 down- and up-regulated canonical pathways in livers from LKO vs. control mice. **d** Mice were treated as described in Fig. 1e and liver lysates from refed mice were immunoblotted as indicated (*n* = 4 mice/group). **e, f** Primary hepatocytes were isolated from control and LKO mice and (**e**) expression of the indicated genes was determined by qPCR (*n* = 5 independent mice/group), or (**f**) immunoblotted as indicated (*n* = 4 independent mice/group). *p*-value: Spring (0.0098); Hmgcr (0.0186); Sqle (0.0098) Scd1 (0.034). **e** Each bar and error represent the mean ± SEM. \**p* < 0.05, \*\**p* < 0.01 analyzed by (**e**) two-way ANOVA with Holm-Sidak post hoc analysis. Source data are provided as a Source Data file.

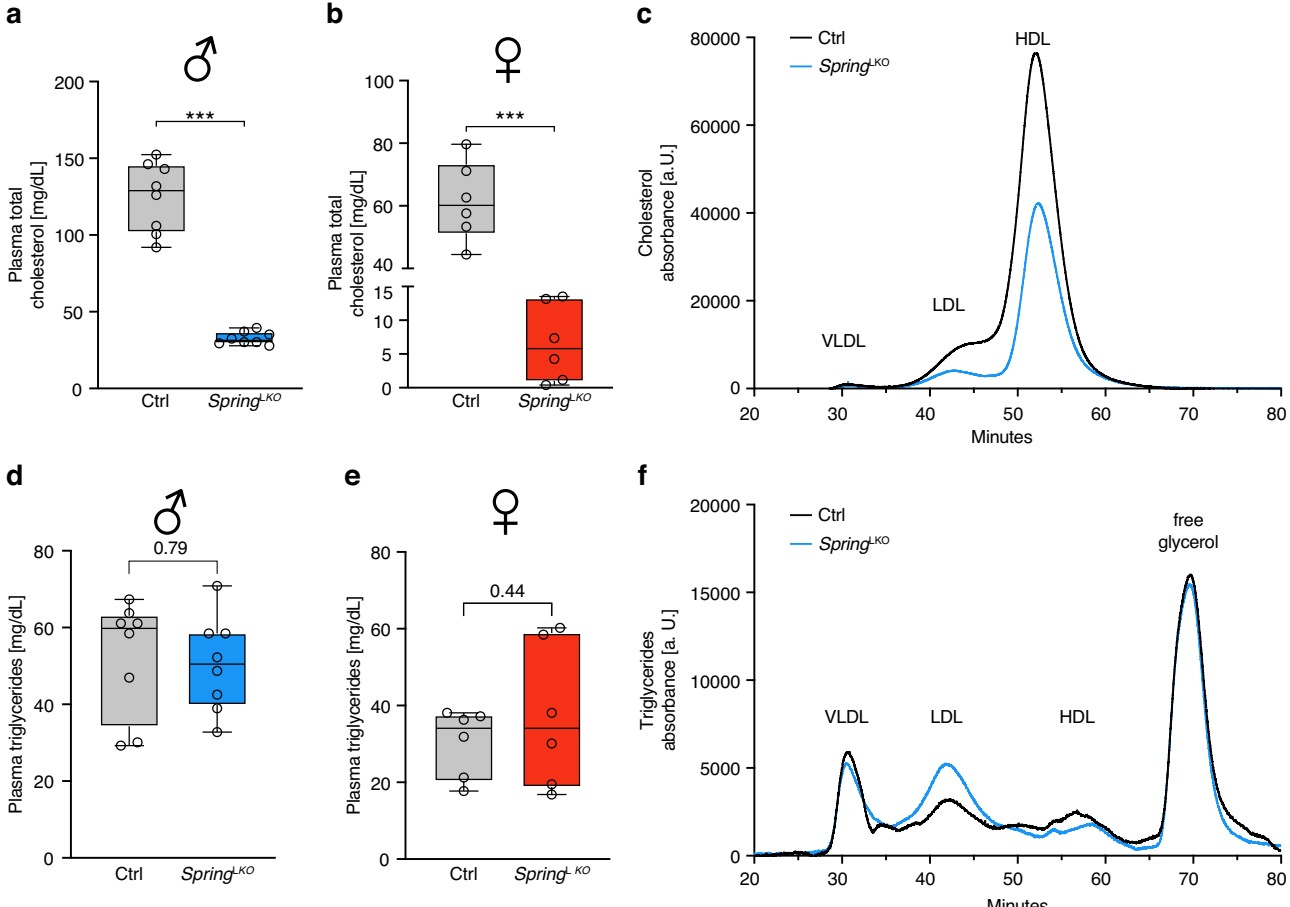

**Fig. 3 | LKO mice have altered plasma lipid levels and lipoprotein profile.** Plasma was collected from 8-week-old control and LKO (**a, d, c, f**) male (*n* = 8 mice/group) and (**b, e**) female mice (*n* = 6 mice/group). The level of (**a, b**) total cholesterol (*p*-value male and female <0.0001) and (**d, e**) triglycerides was determined as shown. **c, f** Plasma was collected from control and LKO male mice (*n* = 8/group) and divided to 4 pools (*n* = 2/pool). Each pool was fractionated individually and the (**c**) cholesterol and (**f**) triglyceride content was continuously measured. The plotted line depicts the average cholesterol or triglyceride levels. (**a, b, d, e**) Box plots show the median (middle line), 25th, 75th percentile (box) and minimum and maximum values (whiskers). ****p* < 0.001 analyzed by (**a, b, d, e**) two-tailed Welch's *t*-test. Source data are provided as a Source Data file.

lethality[29], while that of *Srebf2* (encoding SREBP2) is not tolerated[30]. Similarly, constitutive deletion of either *Mbtps1*[21], *Scap*[31], or *Spring*[10] leads to in utero lethality. This is not surprising given the heavy metabolic demand for lipids to support rapid gestational growth, which cannot be adequately matched in the absence of an intact SREBP pathway. Since SPRING is implicated in regulating SREBPs via SCAP- and S1P-dependent processes, a comparison of the consequences from their hepatic loss is enlightening[10,21,31,32]. In the liver (i) cholesterol levels are lower in *Scap* and *Spring* mice, but unchanged in *Mbtps1* mice, and (ii) triglyceride levels are lower in all three. In plasma, (iii) total cholesterol and triglycerides are lower in all mice with the only exception being that the latter is largely unchanged in *Spring* LKO. Moreover, (iv) ablation of all 3 genes results in decreased cholesterol synthesis in isolated primary hepatocytes, and (v) uniformly reduced the expression of *Srebf1* and *Srebf2* target genes.

This comparison illustrates the largely overlapping consequence of hepatic loss of *Scap*, *Mbtps1*, and *Spring*, consistent with the latter's proposed mechanism of action. In this light, our study and that by Xiao et al. recently demonstrated that *Spring* enhances the proteolytic maturation of S1P (encoded by *Mbtps1*)[13,14]. However, our study reported that despite attaining its mature form and Golgi localization, albeit both with lower efficiency, forced expression of S1P in Hap1-*Spring*[KO] cells was unable to restore SREBP signaling[14]. This contrasts

with the restoration of SREBP function in these cells when SCAP is introduced[10]. Moreover, if enabling the maturation of hepatic S1P would be the primary function of SPRING, one would anticipate identifying changes in other established S1P-regulated pathways[7]. This possibility was not extensively evaluated in the current study. However, the global transcriptomic and proteomic analysis of livers from LKO mice did not identify such alterations, and the expression level of ATF6- and CREB3L3-regulated targets in livers from fed or fasted LKO mice were similar to that of controls (Supplementary Fig. 5). Additionally, no immediate changes in lysosomes were observed in electron microscope images of livers from LKO mice, as would have been expected due to impaired S1P-mediated cleavage of GNMTB[18]. Future studies addressing the relative role(s) of S1P in the SPRING-dependent regulation of hepatic SREBP function, and of other S1P substrates, are therefore highly warranted.

The findings of reduced total cholesterol - observed primarily in the high-density lipoprotein (HDL) and also in the low-density lipoprotein (LDL) fraction - in plasma of LKO mice provides a plausible explanation for the identified genetic signals for HDL and ApoA1 in the *SPRING* locus. The identified SNP (*rs10507274*-C) results in a p.Gln55Arg change in SPRING (SPRING$_{Q55R}$), which is associated with elevated circulating HDL and ApoA1 levels in humans. The mechanism linking SPRING p.Gln55Arg to increased HDL in humans is unclear. Functional prediction algorithms (8 of 10), including SIFT[33] and

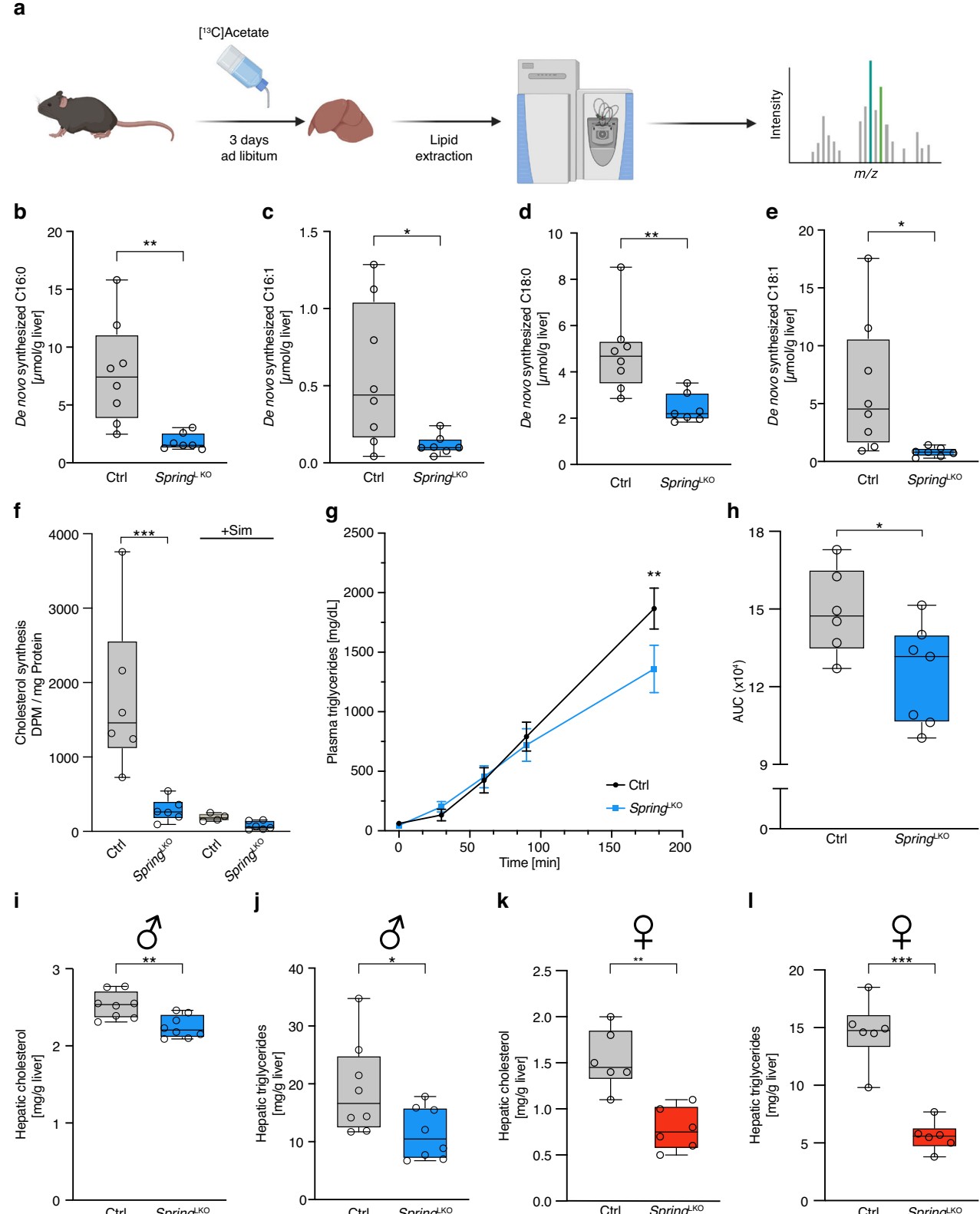

PolyPhen[34], considered this change to be benign and tolerated, with only Mutation taster[35] and LRT[36] scoring this change as deleterious. The variant amino acid is located distal to the S1P cleavage site in SPRING we identified (R$_{45}$NLL$_{48}$), and is not expected to preclude the cleavage event or their interaction (Supplementary Fig. 6a)[14]. The atomic structure of SPRING alone or in complex with S1P has not been reported to date. We therefore used AlphaFold2[37] for structure

prediction of the SPRING-S1P complex (Supplementary Fig. 6b). We analyzed 5 separate models of mutant (SPRING$_{R55}$) and wild type (SPRING$_{Q55}$) using UCSF Chimera[38]. In 3 out of 5 mutant model predictions we observed a steric clash between Asp52 and Arg55 (Supplementary Fig. 6c). Additionally, we observed the loss of a hydrogen bond between His53 and Gln55, which is present in wild type SPRING models (Supplementary Fig. 6c). No steric clashes were found in this

**Fig. 4 | Decreased de novo lipogenesis in LKO mice. a** Scheme depicting the experimental design used to measure in vivo DNL. Male, 8-week-old control ($n = 8$) and LKO mice ($n = 7$) were given ad libitum access to water containing [$^{13}$C]acetate for 3 days as described in the "Methods" section. Liver was collected and the incorporation of [$^{13}$C]acetate into the (**b**–**e**) indicated fatty acids was determined. *p*-values: C16:0 (0.006); C16:1 (0.029); C18:0 (0.002); C18:1 (0.019). **f** Primary hepatocytes were isolated from control and LKO mice. Incorporation of [$^{14}$C]acetate into cholesterol was determined as described without or with 5 μM simvastatin and 100 μM mevalonate to inhibit HMGCR (three independent experiments done with two mice each; in total $n = 6$ mice/group). *p*-value: 0.0004. **g** Male, 8-week-old control ($n = 6$) and LKO ($n = 7$) mice were injected with poloxamer 407 to inhibit peripheral LPL activity as described in the methods section. Subsequently, plasma was collected at the indicated times and the triglyceride level determined. *p*-value: 0.0023. **h** The area under the curve of triglycerides over time is plotted. *p*-value: 0.034 (**i**–**l**) Hepatic content of (**i**, **k**) cholesterol (*p*-value: male (0.0035); female (0.0012)) and (**j**, **l**) triglycerides (*p*-value: male (0.037); female (0.0002)) was determined in (**i**, **j**) male and (**k**, **l**) female control and LKO mice (male: $n = 8$/group; female $n = 6$/group). **g** Each point depicts the mean ± SD. **b**–**f**, **h**–**l** Box plots show the median (middle line), 25th, 75th percentile (box) and minimum and maximum values (whiskers). \**p* < 0.05, \*\**p* < 0.01, \*\*\**p* < 0.001 analyzed by (**b**–**e**, **h**–**l**) two-tailed Welch's *t*-test, (**f**) one-way ANOVA with Holm-Sidak post hoc analysis, or (**g**) two-way ANOVA with Holm-Sidak post hoc analysis. Source data are provided as a Source Data file.

region in all wild type models of SPRING (Supplementary Fig. 6d). These changes could influence the flexibility of this unstructured coiled region and thereby the positioning of the SPRING cysteine-rich domain[10] (Supplementary Fig. 6a), which is both necessary and sufficient for interacting with S1P[12–14] and SCAP[10], and for promoting S1P proteolytic maturation and restoration of SREBP signaling in SPRING[KO] cells[14]. Hence, it is possible that the the p.Gln55Arg mutation may influence the strength and/or affinity of the interaction between the SPRING cysteine-rich domain and S1P, and thereby modulate their function in the SREBP pathway.

How this speculative mechanism could affect circulating HDL levels in humans is not intuitive. Intriguingly, with the advent of whole exome sequencing technology three individuals carrying severe homozygous loss-of-function *MBTPS1* alleles have been recently reported[39–41]. Clinically, these individuals present with skeletal dysplasia and elevated levels of mistargeted lysosomal enzymes in their circulation. One of these studies also reported on the level of circulating lipids. Remarkably, despite having only ~1% functional *MBTPS1* transcripts, plasma lipid levels were largely unchanged in the patient as compared to the unaffected mother and sister[39], the only notable difference being a reduction in plasma HDL levels. This may point towards S1P activity being a determinant of circulating HDL levels in humans. While speculative, differential effects on S1P activity by SPRING variants may thus provide a potential mechanism linking SPRING function to plasma HDL levels.

Finally, pharmacological or antisense-mediated inhibition of hepatic SCAP function in rodents and non-human primates is emerging as a potential lipid-lowering and hepato/cardio-protective strategy[32,42,43]. While early, it is noteworthy to mention that the consequence of loss of hepatic Spring bears semblance to that seen when SCAP activity is reduced. Namely, this improves the plasma lipid profile, reduces hepatic lipid content, decreases fasting glucose, and protects mice from fructose-induced hepatosteatosis; all with no measurable or overt signs of hepatic damage. It is therefore compelling to consider the therapeutic potential of targeting hepatic SPRING in lipid-associated metabolic diseases.

## Methods

### Ethical statement
All the described animal experiments in this study received approval from the Institutional Ethical Committee on Animal Experimentation of the Amsterdam UMC and adhere to national guidelines.

### Materials
Reagents used in this study are listed in Supplementary Table 1. All other standard chemicals were obtained from Sigma.

### Isolation and culture of primary hepatocytes
Primary mouse hepatocytes were isolated using a two-step perfusion method as previously described by Gilglioni et al.[44]. Briefly, livers were cannulated via the portal vein and perfused at a flow rate of 6 ml/min with calcium-free Krebs buffer supplemented with 23.9 mM NaHCO₃

and 10 mM HEPES at 37 °C for 10 min. Subsequently, the buffer was supplemented with collagenase type IV (100 U/ml, Merck) and 2.3 mM CaCl₂ for 10 min. All perfusion buffers were saturated with O₂/CO₂ (95%:5%) to ensure proper oxygenation. The digested liver was transferred to a sterile cell culture dish and submerged in calcium-containing Krebs buffer. Subsequently, hepatic cells were mechanically dispersed and filtered through a 70 μm sieve to remove cell aggregates. The cells were then centrifuged at 50 g for 2 min at 4 °C and washed 3 times in calcium-containing Krebs buffer supplemented with 0.2% bovine serum albumin (BSA). Washed cells were resuspended in warm Dulbecco's modified Eagle's media (DMEM) supplemented with glucose (1 g/L), 26 mM sodium bicarbonate, 100 nM dexamethasone (Sigma), 100 nM insulin (Gibco), penicillin (100 units/mL) and streptomycin (100 μg/mL), and cell number and viability determined by trypan blue exclusion. For experiments, cells were seeded in collagen I coated cell culture dishes (Corning) and after adherence the medium was supplemented with 10% fetal bovine serum.

### Mouse experiments
The generation of *Spring*[(fl/fl)] mice has been described in Loregger et al.[10]. Expression of Cre-recombinase results in excision of exons 2-5. Alb-Cre⁺ mice (a kind gift from Dr. R. Oude Elferink, AMC) were crossed with *Spring*[(fl/fl)] to obtain liver-specific *Spring* knockout mice (LKO). Mice were maintained by heterozygous mating of Alb-Cre⁺-*Spring*[(fl/fl)] (LKO) with Alb-Cre⁻*Spring*[(fl/fl)] (control) mice and in all experiments Alb-Cre⁻ *Spring*[(fl/fl)] littermates were used as controls. Rosa26-Cre[ERT2] mice (a kind gift from Dr. Ivo Huijbers, NKI) were similarly crossed with *Spring*[(fl/fl)] and maintained as above. Phenotypic characterization: 10–12-week-old control and LKO male mice ($n = 8$/group) were monitored for 10 week. Mice were weighed at the start of this period and repeatedly at weekly intervals. To determine plasma lipid and glucose levels mice were fasted for 4 h (13:00) and blood collected via a tail incision ($t = 0,4,8$ weeks). After 10 weeks mice were killed. Blood was obtained by retro-orbital puncture. Tissues were snap frozen in liquid nitrogen and stored at −80 °C for subsequent analysis. Livers were also fixed in phosphate buffered 4% paraformaldehyde or embedded in TissueTek (Sakura Finetek) followed by snap freezing at −80 °C in isopentane for histological analysis or Oil Red O staining, respectively. The oil red O positive area was determined using ImageJ2 (V2.9.0/1.53t). Glucose tolerance test: 12-week-old mice (as above) were fasted for 4 h. Subsequently, mice were administered 2 g per Kg glucose (Sigma) via intraperitoneal injection. Tail vein blood was sampled at the indicated time points and blood glucose measured. Insulin tolerance test: 16-week-old mice (as above) were fasted for 4 h followed by an intraperitoneal injection of 1 U per Kg insulin (Gibco). Blood was sampled as described above. Fasting-refeeding challenge: To evaluate the hepatic post-prandial response 17–20-week-old male LKO ($n = 18$) or control mice ($n = 17$) were either fasted for 20 h (LKO $n = 9$; control $n = 8$), or fasted for 16 h and then refed for 4 h (LKO $n = 8$; control $n = 9$). Subsequently, mice were killed and livers collected as described above for downstream analysis. High-fructose diet challenge: 7–10-week-old male LKO ($n = 9$) or control mice ($n = 6$) were fed a HFrD (TD.89247, Envigo) for 10 weeks. Mice were

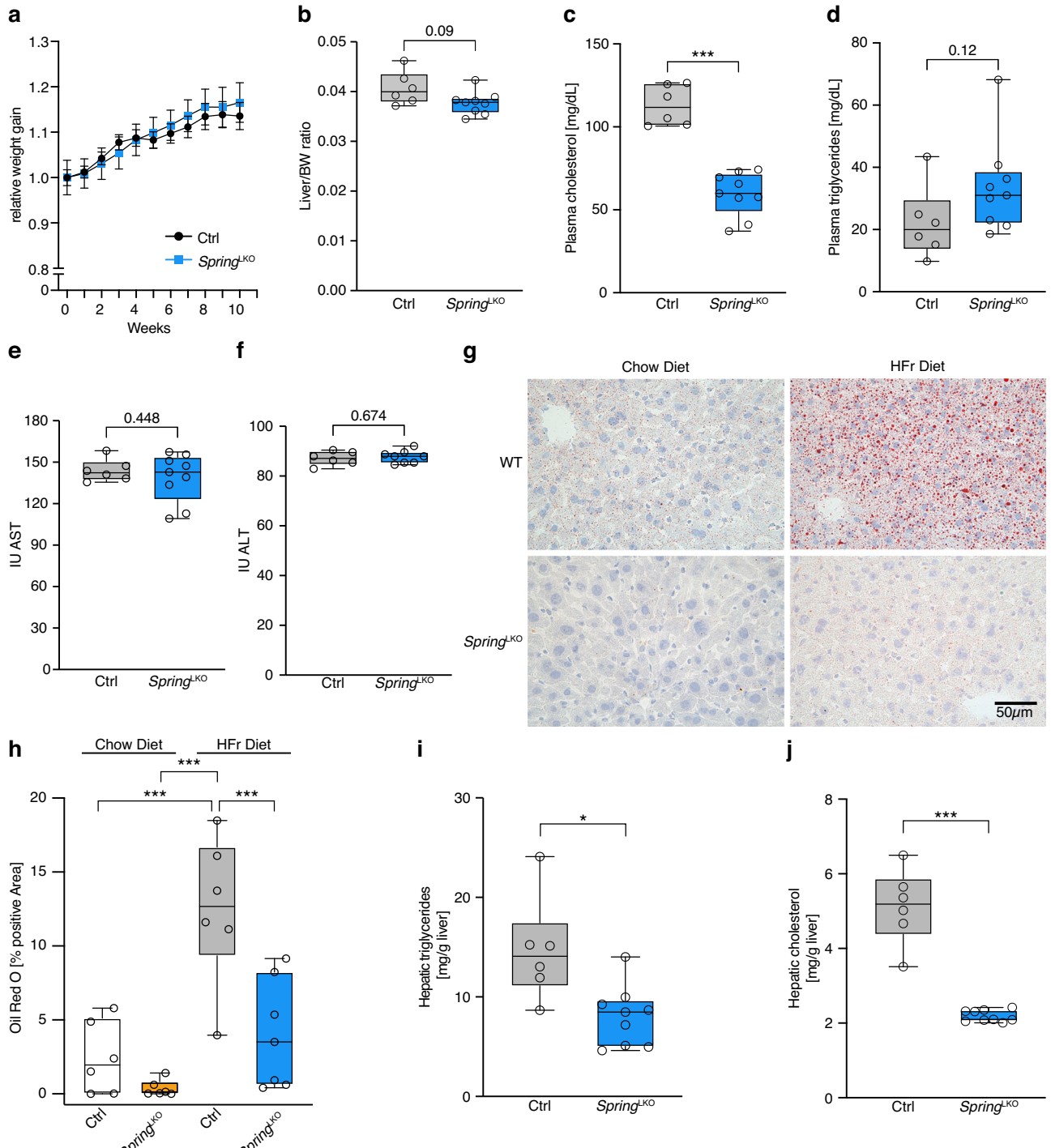

**Fig. 5 | LKO mice are protected from fructose-induced hepatosteatosis.** Male, 8-week-old control ($n = 6$) and LKO ($n = 9$) mice were fed a fructose diet for 10 weeks. **a** The change in body weight over the time course of the study, and of the (**b**) liver/body weight ratio at killing are plotted. **c–f** At killing (10 weeks), plasma was collected and the level of (**c**) cholesterol, (**d**) triglycerides, (**e**) AST, and (**f**) ALT were determined. **g, h** Hepatic neutral lipids were stained with Oil-red-O. **g** A representative image from chow- and fructose fed control or LKO liver, and (**h**) quantification of images from independent mice are shown. **i, j** Hepatic content of (**i**) triglycerides and (**j**) cholesterol from control and LKO mice fed a fructose-rich diet for 10 weeks. **a** Each point and error represent the mean ± SD. **b–f, h–j** Box plots show the median (middle line), 25th, 75th percentile (box) and minimum and maximum values (whiskers). *$p < 0.05$, ***$p < 0.001$ analyzed by (**a**) two-way ANOVA with Holm-Sidak post hoc analysis, (**h**) one-way ANOVA with Holm-Sidak post hoc analysis, or (**b–f, i, j**) two-tailed Welch's $t$-test. Source data are provided as a Source Data file.

weighed weekly and after 4, 8, and 10 weeks blood was sampled and glucose measured as described above. In vivo hepatic VLDL production assay: 7–10-weeks-old male LKO ($n = 7$) or control mice ($n = 6$) were fasted for 4 h and subsequently received an intraperitoneal injection of poloxamer 407 (1 mg/g mouse; Merck) to inhibit LPL activity. Blood was sampled at the indicated time points to determine triglyceride levels. Inducible global deletion of Spring in adult mice: 14–16-week-old male Spring$^{(fl/fl)}$ $^{CreERT2+}$ ($n = 5$) or Spring$^{(fl/fl)}$ $^{CreERT2-}$ ($n = 6$) mice were

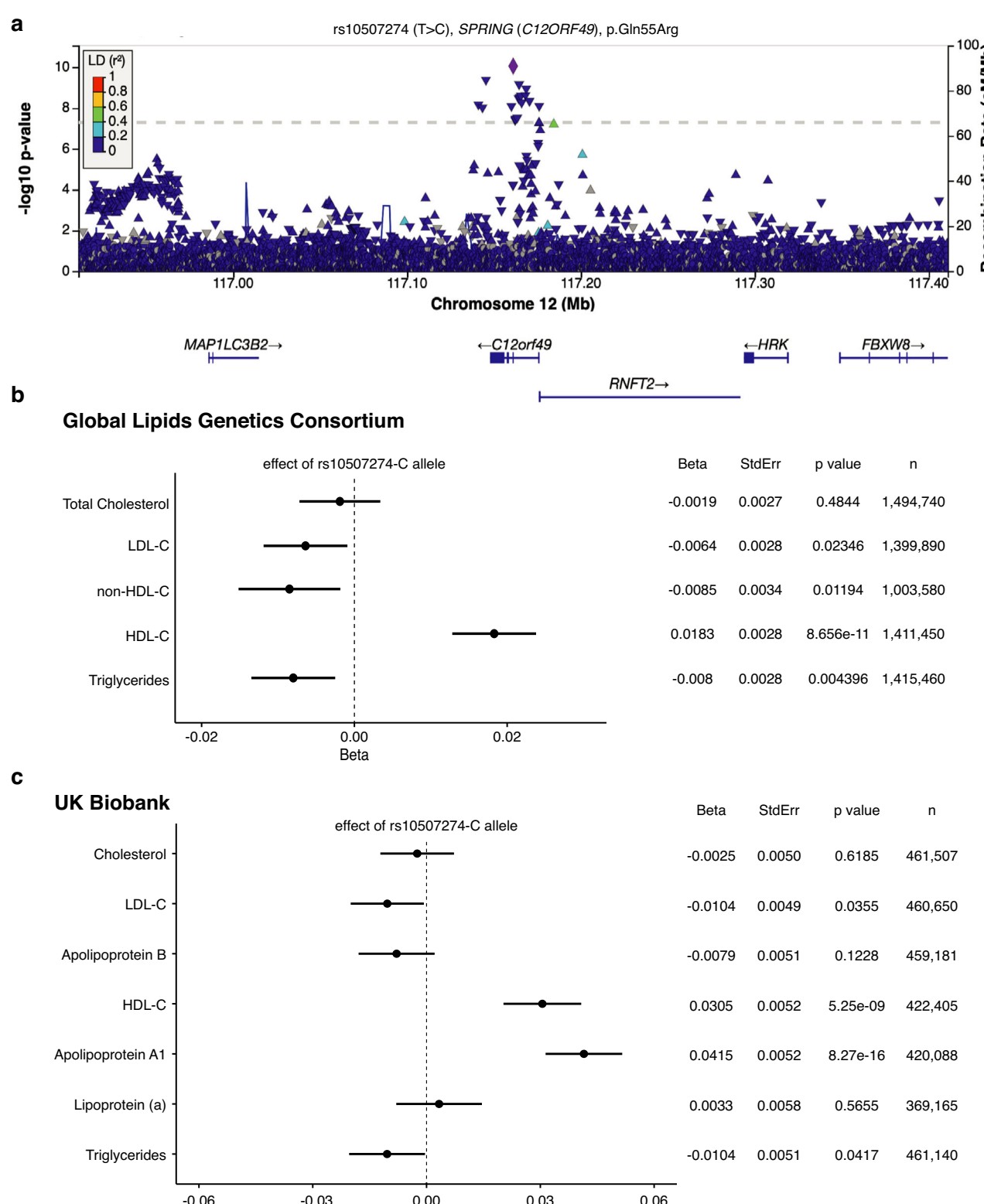

**Fig. 6 | Genetic associations of variants in the *SPRING* gene locus with plasma lipids and biomarkers. a** Locus zoom of genetic associations of variants in the *SPRING* locus (12q24.22) with HDL cholesterol. Each variant identified in the 1000 genomes project is represented by a triangle, with color representing linkage disequilibrium (LD) with the lead SNP (rs10507274, purple diamond). A single variant, rs10507274, demonstrates the most significant association with HDL cholesterol plasma levels (*p* = 8.656e-11). **b** Genetic associations of rs10507274 with plasma lipids and lipoproteins in the Global Lipids Genetics Consortium dataset. Effects on circulating total cholesterol, LDL-C, non-HDL-C, HDL-C, and triglycerides are

plotted per rs10507274-C allele. Each point and error represent the beta value ± SE. Corresponding beta, standard error *p* values, and number of individuals are shown on the right. **c** Genetic associations of rs10507274 with plasma lipids, lipoproteins and apolipoproteins in the UK-Biobank. Effects on circulating total cholesterol, LDL-C, HDL-C, triglycerides, Apolipoprotein A1, B, and lipoprotein (a) are plotted per rs10507274-C allele. Each point and error represent the beta value ± SE. Corresponding beta, standard error, *p* values and number of individuals are shown on the right.

interperitoneally injected with Tamoxifen (120 mg/kg mouse; Sigma) on 5 consecutive days. By 2 weeks post the last injection Spring[(fl/fl) CreERT2+] mice reached humane endpoint criteria requiring their killing thereby precluding detailed analysis of this model. The mice used in this study had a C57BL6/J genetic background.

## Immunoblot analysis and co-immunoprecipitation
Tissues were homogenized for 1 min at a frequency of 30 Hz using a TissueLyser II (Qiagen). Total cell lysates or tissue homogenates were prepared for immunoblotting in radio-immunoprecipitation assay (RIPA) buffer (Boston Biochem), which was supplemented with 1 mM phenylmethylsulfonyl fluoride (Sigma) and protease inhibitors (Roche) by agitation at 4 °C for 30 min. Subsequently, lysates were cleared by centrifugation at 4 °C at 10,000×g for 10 min. The protein concentration of the cleared lysates was determined using a BCA assay (ThermoFisher) following the manufacturers' protocol, and an equal amount was loaded for analysis. Samples were separated on NuPAGE Novex 4-12% Bis-Tris gels (ThermoFisher) and transferred to nitro-cellulose membranes, blocked in 5% milk (Elk) in phosphate-buffered saline (PBS) supplemented with Tween and subsequently probed with primary antibodies. Primary antibodies used in this study are listed in Supplementary Table 2. Secondary HRP-conjugated antibodies (A28177 & A27036, Invitrogen) were used and visualized with chemiluminescence on an IQ800 (GE Healthcare), and quantified using ImageJ. Unless indicated, immunoblots are representative of at least three independent experiments with similar results.

## Quantitative PCR
RNA was isolated using the Direct-zol RNA miniprep kit (Zymo Research) following the manufacturer's protocol. cDNA was generated using the iScript reverse transcription reagent (BioRad). SensiFAST SYBRgreen (Bioline) was used for real-time quantitative PCR (RT-qPCR) on a LightCycler 480 II system (Roche). Gene expression was normalized to the expression of 36B4 and GAPDH. Primer sequences are shown in Supplementary Table 3.

## Measurement of hepatic de-novo lipogenesis in LKO mice
Hepatic de-novo lipogenesis (DNL) was measured as we previously reported[45]. Briefly, the indicated mice received 2% [1-13C]acetate (Sigma) in their drinking water starting 3 days before killing. Hepatic lipids were extracted from homogenates (10% w/w in PBS) according to Bligh and Dyer[46]. Fatty acids were then methylated by incubating the lipid extracts in a 5:1 (v/v) mixture of methanol/HCL (6N) for 4 h at 90 °C and extracted twice using hexane. Isotopomer distributions of the C16:0, C16:1, C18:0, and C18:1 fatty acid methyl esters were then analyzed by GC/MS (Agilent 5975 C GC/MSD, Agilent Technologies, Santa Clara, CA) using ammonia-induced chemical-ionization (CI). The measured isotopomer distribution patterns were then corrected for the natural abundance of 13C to obtain the excess fractional distribution of the isotopomers ($m_{+0} - m_{+x}$) due to incorporation of the labeled 13C-acetate into the fatty acid molecules[47]. Mass isotopomer distribution analysis was then used to calculate fatty acid synthesis, as previously described[45].

## Proteomics analysis of liver tissue
Liver tissue from control and LKO mice ($n = 6$/group) were homogenized in 8 M urea, Tris-HCl 0.1 M pH 8.5 buffer supplemented with protease inhibitors (1:100, Cell Signaling). Homogenized samples were incubated at 4 °C for 60 min with constant shaking and then cleared for 30 min at 14,000×g at 4 °C. The supernatant containing the extracted proteins was collected and quantified using the Lowry protein assay. Subsequently, a volume corresponding to 10 µg of protein was fully dried using a vacuum concentrator at 45 °C for 45 min. The dried protein pellet was resuspended in 10 µl of water with the addition of 10 µl of Ammonium bicarbonate solution 50 mM (Final pH 8.5).

Proteins were reduced for 30 min at 55 °C incubation with 5 mM DTT, and then alkylated at room temperature for 30 min in the dark with 150 mM iodoacetamide in ammonium bicarbonate solution 50 mM (15 mM final concentration) at room temperature for 30 min in the dark. Trypsin digestion (Merck, 1:20 enzyme to protein ratio) was carried out overnight at 37 °C and was terminated by acidification with 1% trifluoroacetic acid. Samples were analyzed using a Dionex Ultimate 3000 nano-LC system (Sunnyvale CA, USA) connected to an orbitrap Fusion™ Tribrid™ Mass Spectrometer (Thermo Scientific, Bremen, Germany) equipped with a nano-electrospray ion source operating in positive ion mode. The peptide mixtures were pre-concentrated onto an Acclaim PepMap C18 5 µm, 100 Å, 100 µm ID × 2 cm (Thermo Scientific) and separated at 35 °C on an EASY-Spray PepMap RSLC C18 column: 3 µm, 100 Å, 75 µm ID × 25 cm (Thermo Scientific), using mobile phase A (0.1% aqueous formic acid) and mobile phase B (0.1% aqueous formic acid /acetonitrile (2:8)) at a flow rate of 300 nL/min. MS spectra were collected over an $m/z$ range of 375–1500 Da at resolution 120,000 in the data dependent mode, cycle time 3 sec between master scans. Fragmentation was induced by higher energy collisional dissociation (HCD) with collision energy set at 35 eV. The raw MS data files were converted to mzML format (centroid mode) using the MSconvert tool of the software ProteoWizard (version 3.0.1957). Files (mzML format) were then analyzed using an OpenMS ver. 2.4 nodes operating within the open-source software platform KNIME® ver. 4.1.1. Peptides were identified using the previously described peptides identification approach that combined search engines[48]. The OpenMS PeptideIndexer node was used to index peptide sequences, with leucine/isoleucine equivalence set. Protein inference was then carried out using the Protein Inference Analysis (PIA) algorithm. Protein abundance estimates were calculated using the node Feature-FinderMultiplex prior generation of spectral features, followed by PIA-assisted FDR-multiple score estimation and filtering (combined FDR score 0.01), their ID mapping and combination with peptide IDs, and their subsequent alignment, grouping, and normalization (eMapAlignerIdentification, FeatureUnlabeledQT and ConsensusmapNormalizer nodes). Proteins and peptides label free quantification (LFQ) was then processed with the OpenMS ProteinQuantifier node based on intensities of the most abundant identified peptides ($n = 3$). The corresponding output files were read as tables from the CSVreader node output. For the functional analysis of the proteome, proteins showing significant changes were evaluated using QIAGEN's Ingenuity® Pathway Analysis software (IPA®, QIAGEN).

## RNAseq analysis of liver tissue
RNA from liver tissue was isolated with the Direct-zol RNA miniprep kit (Zymo Research) and quantified fluorometrically with a Qubit 3 Fluormeter (Invitrogen). The Quality of the RNA was evaluated with a 2100 Bioanalyzer (Agilent Technologies). A KAPA mRNA hyperPrep kit (KAPA Biosystems) was used for mRNA library construction and sequencing was performed on an Illumina NovaSeq 6000 at the Core Facility genomics of the Amsterdam UMC, location AMC, Amsterdam, the Netherlands. Bioinformatic analysis was performed in collaboration with the Bioinformatics Core Unit of the Amsterdam UMC, location AMC, Amsterdam, the Netherlands as previously described[49]. Briefly, quality control (fastQC, dupRadar, Picard Tools)[50] was performed on the reads. Subsequently they were trimmed using Trimmomatic v0.32[51] and aligned to the mouse genome using HISAT2 (v2.1.0)[52]. Counts were obtained using corresponding GTF and HTSeq (v0.11.0)[53]. TMM (EdgeR)[54] and limma/voomR[55] packages were used to perform statistical analysis. Count data was transformed to log2-counts per million (logCPM) and normalized using the trimmed mean of M-values method[54]. Voom was used to precision weigh the data. Bayes moderated t-test was used to assess differential expression within limma's linear model framework and the precision weights estimated by voom[55]. The mouse ID was included as a random effect in

the linear model to estimate the consensus correlation, using duplicateCorrelation from the limma package. The p values were corrected for multiple testing using the Benjamini-Hochberg false discovery rate. BiomaRt (release 94) was used to reannotate the genes. Results are shown in an in-house made Shiny-app. Analysis was performed using R v3.5.0 and Bioconductor v3.7.

## Analysis of plasma
Aspartate aminotransferase (AST) (92025, AST TGO Colorimetric Method, Biolabo) and alanine transaminase (ALT) (92027, ALT TGP Colorimetric Method, Biolabo) levels were determined in plasma samples following manufacturers' protocols. Blood glucose was measured with a handheld glucometer (Contour XT, Ascensia, Bayer). Concentrations of circulating lipids were measured using commercially available enzymatic kits for triglycerides (TG, Diagnostic Systems) and total cholesterol (TC; Biolabo).

## FPLC analysis of plasma lipoproteins
Total cholesterol (TC) and triglyceride (TG) content in the main lipoprotein classes (VLDL, LDL and HDL) was determined using fast protein liquid chromatography (FPLC), as previously described[56]. The main system consisted of a PU-4180 quaternary pump with an DG-4000 in-line degasser and a Diode Array Detector (DAD MD 4015, Jasco). After injection of 30 μl plasma (1:1 diluted with TBS) the lipoproteins were separated with a Superose 6 increase 10/30 column (GE Healthcare). As eluent Tris buffered Saline (pH 7.4) was used at a flow rate of 0.31 ml/min. An auxillary (PU-4080i Plus, Jasco) was used for either in-line cholesterol PAP or Triglyceride enzymatic substrate reagent (Sopachem) addition at a flowrate of 0.1 ml/min facilitating TC or TC detection at 505 nm. Commercially available lipid plasma standards (low, medium and high) were used for generation of TC or TG calibration curves for the quantitative analysis (SKZL) of the separated lipoprotein fractions. All chromatographic analysis was carried out using Chrom Nav chromatographic software, version 2.0 (Jasco).

## Measurement of cholesterol synthesis in primary hepatocytes
Determination of cholesterol synthesis in cells was done as previously described in Loregger et al.[57]. Briefly, primary hepatocytes were isolated as described above and cultured in Dulbecco's modified Eagle's media (DMEM) supplemented with glucose (1 g/L), 26 mM sodium bicarbonate, 100 nM dexamethasone, 100 nM insulin, penicillin (100 units/mL), streptomycin (100 μg/mL), 10% fetal bovine serum and 0.1 μCi/ml [$^{14}$C]-acetate (American Radiolabeled Chemicals) for 16 h in the presence or absence of 5 μM simvastatin and 100 μM mevalonate, as indicated. Cells were washed twice with Hank's balanced salt solution (HBSS; Gibco) and lysed in 0.1 M NaOH. The non-saponifiable lipid fraction was subjected to scintillation counting to determine radioactive content and normalized to the protein content as measured with the BCA assay. Cholesterol synthesis is expressed as DPM / mg cellular protein.

## Genetic association of variants in the SPRING locus with plasma lipids
The Global Lipids Genetics Consortium aggregated GWAS results from 1,654,960 individuals from 201 primary studies representing five genetic ancestry groups[28]. Summary statistics from the meta-analysis results were downloaded via the following link http://csg.sph.umich.edu/willer/public/glgc-lipids2021. We screened a genomic region 250 kb up- and downstream of the SPRING gene.

## Genetic association with circulating biomarkers in the UK Biobank study population
The UK Biobank study, described in detail previously[58], is a population-based prospective cohort in the United Kingdom in which >500,000 individuals aged between 40 and 69 years were included from 2006 to 2010. The study has been approved by the North West Multi-Centre

Research Ethics Committee for the United Kingdom, from the National Information Governance Board for Health and Social Care for England and Wales, and by the Community Health Index Advisory Group for Scotland (https://www.ukbiobank.ac.uk/media/0xsbmfmw/egf.pdf) and all participants have given informed consent (https://www.ukbiobank.ac.uk/media/gnkeyh2q/study-rationale.pdf). Blood biomarker measurement details are described in length in the UK-biobank showcase (https://biobank.ndph.ox.ac.uk/showcase/label.cgi?id=17518). Briefly, plasma cholesterol, triglyceride and glucose plasma levels were measured using enzymatic assays (https://biobank.ndph.ox.ac.uk/showcase/showcase/docs/serum_biochemistry.pdf). High-density lipoprotein Cholesterol (HDL-C) was measured using an enzyme immuno-inhibition method while low density lipoprotein cholesterol (LDL-C) was measured using an enzymatic selective protection method. Apolipoprotein A1 (apoA1), Apolipoprotein B (apoB), High Sensitivity C-Reactive Protein (CRP), and Lipoprotein (a) Lp(a) were measured using immuno-turbidimetric assays (Beckman Coulter; Lp(a): Randox Bioscience). Alkaline phosphatase (ALP), Alanine aminotransferase (ALT), aspartate aminotransferase (AST), and gamma-glutamyltransferase (GGT) were measured using enzymatic rate method. All biomarkers except glycated haemoglobin (HbA1c) were measured on a Beckman Coulter AU5800 (Beckman Coulter). HbA1c was measured in mmol/mol (https://biobank.ndph.ox.ac.uk/showcase/showcase/docs/serum_hb1ac.pdf) with high performance liquid chromatography method (Bio-Rad Laboratories) on Bio-Rad Variant II Turbo analyzers.

## Genetic association of common variants with biomarkers in the UK Biobank
The genetic associations with biomarkers were extracted from the Pan-ancestry genetic analysis of the UK Biobank, by the Pan-UK Biobank team [released June 15, 2020]. In short, genetic and phenotypic data from ~500,000 participants in the UK Biobank (https://www.ukbiob4ank.ac.uk) were used to conduct a Genome-Wide Association Study (GWAS). Genotypes were imputed from the Haplotype Reference Consortium plus UK10K & 1000 Genomes reference panels as released by UK-Biobank in March 2018. This research has been conducted using the UK Biobank Resource (project ID 31063), and the use of these data is bound by all terms of usage of the UK Biobank. A p value below 5.0E-8 was considered as genome-wide significant. The functional consequence of the identified SPRING coding variant rs10507274 (resulting in a p.Gln55Arg change) was evaluated using the following prediction algorithms: SIFT[33], Polyphen2-HDIV[34], Polyphen2-HVAR[34], LRT[36], Mutation Taster[35], Mutation Assessor[59], FATHMM[60], PROVEAN[61], MetaSVM[62], and MetaLR[62].

## Whole-exome sequencing
Full details of the whole-exome sequencing (WES) in the UK Biobank have been reported previously[63]. In short, WES was performed using IDT xGen Exome Research Panel v1.0, targeting 38 997 831 bases in 19 396 genes. Exomes were captured including 100 bp flanking regions. To screen the exome sequencing dataset, we used the Genebass resource made available to the public[64]. The dataset encompasses 4529 phenotypes with gene-based and single-variant testing across 394,841 individuals with exome sequence data from the UK Biobank.

## Plotting of human genetic data
Data plotting has been performed using RStudio (v.2022.02.1) with different R-packages reshape2, ggplot2, ggforest and ggridges. Plotting of associations within genetic loci has been performed using LocusZoom[65].

## Statistics
Statistical significance was tested using ANOVA with Holm-Sidak post hoc analysis or t-test with Welch correction. No violation of normal

distribution was found using the Shapiro-Wilk and Kolmogorov-Smirnov tests. Outliers were identified using a ROUT analysis. Prism v9 software was used for statistical analyses and $p < 0.05$ was considered significant. Box plots show the median (middle line), 25th, 75th percentile (box) and minimum and maximum values (whiskers). $p$ values are indicated by asterisks: $*p < 0.05$, $**p < 0.01$, and $***p < 0.001$.

## Reporting summary

Further information on research design is available in the Nature Portfolio Reporting Summary linked to this article.

## Data availability

The raw proteomics data files can be accessed on Figshare [https://figshare.com/articles/dataset/SPRING-dependent_regulation_of_hepatic_SREBP_signaling_governs_systemic_lipid_metabolism_in_mice_and_humans_-_raw_proteomics_files/22699408/3]. The raw RNAseq data files have been deposited in the Gene Expression Omnibus database under accession number GSE236045. All other data supporting the findings of this study are available within the paper and its Supplementary Information. Source data are provided with this paper.

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

## Acknowledgements

The present research has been conducted using summary statistics generated from the Global Lipids Genetics Consortium (http://www.lipidgenetics.org) and from the UK Biobank resource (under applications 26041, 48511, 49823 and 31063). We thank all participants of these cohorts and the research teams for making the data publicly available. We are grateful to the Bioinformatics Core Facility of Nantes BiRD, member of Biogenouest, Institut Français de Bioinformatique (IFB) (ANR-11-INBS-0013) for the use of its resources and for its technical support. The authors would like to thank Prof. Giangiacomo Beretta from the University of Milan for the kind help and assistance in processing the proteomics files in the OpenMS pipeline. NZ is an Established Investigator of the Dutch Heart Foundation (2013T111) and is supported by an ERC Consolidator grant (617376), an ERC Proof-Of-Concept grant from the European Research Council (862537) and by a Vici grant from the Netherlands Organization for Scientific Research (NWO; 016.176.643). We thank members of the Zelcer lab, Marten Hoeksema, Bart v/d Sluis, and Irith Koster for their critical comments and suggestions on this study. GDN is supported by: Progetti di Rilevante Interesse Nazionale [PRIN 2017 K55HLC], Ricerca Finalizzata, Ministry of Health [RF-2019-12370896], PNRR Missione 4, [Progetto CN3 - National Center for Gene Therapy and Drugs based on RNA Technology], PNRR Missione 4, [Progetto MUSA- Multilateral urban sustainability action], PNRR-MAD-2022-12375913. JFdB is supported by the Nutrition & Health initiative of the University of Groningen. AR is supported by a grant funded by the Agence Nationale de la Recherche (ANR-21- CE14-0051) for the GENESIS project. NZ is an Established Investigator of the Dutch Heart Foundation (2013T111) and is supported by an ERC Consolidator grant (617376), an ERC Proof-Of-Concept grant from the European Research Council (862537) and by a Vici grant from the Netherlands Organization for Scientific Research (NWO; 016.176.643).

## Author contributions

S.H., A.L. and N.Z. conceptualized the study. S.H., M.V., J.K., R.O., L.Z. and V.S. conducted experiments. J.H.M.L. measured plasma lipid levels and distribution, M.S. and G.D.N. did the proteomics analysis, A.J. analyzed the transcriptomic data, J.F.dB. and F.K. determined the in vivo lipid synthesis, A.R. analyzed the human genetic data, and K.K. performed the structural analysis. S.H. and N.Z. drafted, edited, and revised the manuscript.

## Competing interests

The authors declare no competing interests.
