## [Peer Review File · Nature Communications]

REVIEWER COMMENTS

Reviewer #1 (Remarks to the Author):

In recent years, the authors and others have identified SPRING/C12orf49 as a regulator of Mbtps1/Site-1 protease maturation. MBTPS1 is a Golgi subtilisin-kexin protease required for multiple pathways. However, it is best known for its role in the proteolytic activation of SREBP family transcription factors. Mechanistic studies indicate that SPRING is required for the autocatalytic proteolytic activation of Mbtps1 and this indirectly impacts SREBP pathway activity. A previous study demonstrated that whole body knockout of SPRING is embryonically lethal in mice.

In this manuscript, Hendrix and colleagues examine the hepatic function of SPRING using Albumin-Cre liver knockout (LKO) mice. The study is very high quality and clearly demonstrates a role for SPRING in regulation of hepatic SREBP activity as expected from cell culture experiments and zebrafish data from others. The study provides strong evidence that SPRING plays an important physiological role in supporting SREBP pathway activity. Importantly, serum cholesterol is dramatically reduced in SPRING LKO mice, indicating that hepatic SPRING may be a potential therapeutic target for dyslipidemia and cardiovascular disease. The authors only see minor effects on expression of other Mbtps1-dependent transcriptional pathway, but animals were not stressed, so additional work is required to conclude whether SPRING plays a role in ATF6 function, for example, in the liver. Overall, the manuscript is very well written, the data support the conclusions and the experimental methods are clear. Below are a few comments that may strengthen the manuscript.

Major:

1. The authors demonstrate that "hepatic SPRING is indispensable for intact SREBP signaling". However, they do not present any experiments to test whether SPRING function is regulated and that this impacts SREBP activity. Thus, the use of the word "regulation" in the title is inappropriate, and the title should be modified to something like "Hepatic SREBP signaling requires SPRING ..."
2. The authors demonstrate that hepatic lipoprotein secretion is decreased in SPRING LKO animals. Given the strong reduction in PCSK9 mRNA and protein, it is also likely that serum cholesterol is reduced due to increased LDLR activity. The authors should either test this directly in mice or at a minimum present it as a likely contributor to the phenotype in the results or discussion.
3. Is there a SPRING antibody that detects hepatic SPRING? If so, levels of SPRING in LKO mice should be tested.

4. In Fig. 2B, please indicate other proteins with significant expression changes. Depending on the identities, please discuss. Supplemental table 5 is not well presented and should better indicate what changes were observed in the MS experiments. What does the score represent? Why is there no fold change column or p value listed?

Minor:

1. Please note the p value for panels in Fig 1D.
2. Figure 1E would be clearer if the X-axis indicated the comparison (LKO/WT).
3. Please address why LXR target genes may be elevated in SPRING LKO (Fig. 2C).
4. Supplemental Tables 4&5 need better titles to explain the data within. Supplemental Table 4 should include the p values for all GO term analyses to confirm that those presented in 2C are the most significant.

Reviewer #2 (Remarks to the Author):

This is a very strong paper that investigates the role of the newly identified SPRING protein in regulating hepatic lipid metabolism. The authors use a newly created liver-specific knockout mouse model to assess the role of SPRING in hepatic de novo lipogenesis (DNL) and cholesterol metabolism, and find that SPRING LKO mice have drastically reduced plasma cholesterol levels in both the HDL and LDL fractions. This appears due in part to a reduction in VLDL secretion caused by reduced hepatic DNL. The authors then attempt to investigate the relationship between SPRING and lipoprotein metabolism in humans through a combined interrogation of the association between common and rare genetic variants and plasma lipid traits in humans. This review focuses on the genetic analysis provided. In summary, the authors present compelling GWAS data and should further investigate said data. Conversely, the missense and loss-of-function variant analysis is vague and speculative and detracts from the overall story. The authors should stay away from speculating about coding variants of unknown consequences.

Specific Comments:

- The GWAS data from the GLGC and the UK Biobank convincingly demonstrates that there is a genetic signal at the C12orf49 locus for HDL-C and ApoA1 levels in humans. This is the strongest piece of genetic evidence provided by the authors, and they should focus their attention here. There are clearly both coding and non-coding variants in this GWAS signal, and due to linkage it is hard to assign causality (identification of the causal variant is likely beyond the scope of this paper). The authors should

investigate eQTL data to see if any non-coding variants associate with gene expression of C12orf49 or other neighboring genes. Any eQTL evidence would strengthen this argument.

- The authors could also complement their AlphaFold analysis with predictive analysis of the consequences of the GWAS missense mutations using predictive tools (SIFT and PolyPhen are two tools but there are others) to see if any of the common variants are predicted to be damaging to SPRING function. This and the approach outlined in the first comment would help strengthen the argument that this GWAS signal is highlighting SPRING as causal in humans.

- While the GWAS data is compelling (but could use some extra support), the missense variant and rare variant analysis is not compelling and somewhat distracting. There is no evidence that Q55R is a gain-of-function mutation. No functional data is provided that demonstrates this in vitro or in vivo. rs10507274 could easily be in linkage disequilibrium with a different coding variant or a non-coding variant that is causal (this goes back to the eQTL analysis). Speculation about the consequence of this missense variant is unwarranted and should be excluded unless the authors want to do the extra work required to prove that this variant is causal and/or gain of function.

- The “definite loss-of-function” variants are also not well substantiated. Two of the variants are in a different isoform of SPRING. The authors do not discuss the relevance of this isoform and thus the LOF variants in this isoform are of unknown significance. Does the short isoform have the same function? Is it expressed in hepatocytes? Do mice have this isoform, and is it knocked out in your LKO? At minimum, the short isoform appears to not have a transmembrane domain suggesting a very different function. The authors should discuss these questions if they want to include those two LOF variants.

- Given that global loss of SPRING in mice is lethal, it would follow that the same is true in humans. Presumably the humans are heterozygous for these LOF variants (the authors should clarify this in the paper), and these authors previously showed that mice with heterozygous global deletion are viable (Loregger 2020). Did those mice have a plasma lipid phenotype? Is there any reason to believe that these humans should have a phenotype if they are heterozygous? Is C12orf49 haploinsufficient?

- Finally, the two LOF variants in the long isoform are near the end of the SPRING protein. Depending on their exact location based relative to exon structure, these variants could escape nonsense mediated decay and be produced as truncated proteins that retain function or are hypomorphic. There’s lots of interesting questions here but they are likely beyond the scope of this paper. Given all the unknowns about these LOF variants, I’d exclude them from the paper because they don’t add to the authors’ conclusions and raise too many further questions.

- Line 519-520 should read “Global Lipids Genetics Consortium.”

Reviewer #3 (Remarks to the Author):

SPRING (C12ORF49) is a recently identified modulator of cholesterol and fatty acid metabolism. Previous studies (largely carried out in cultured cells, references 10-14 of the current paper) by the authors and others have shown that SPRING played an essential role in the activation of SREBPs (Sterol Regulatory Element Binding Protein), key transcriptional activators of genes controlling cholesterol and fatty acid metabolism. In the current paper, the authors continued their in vitro works to study the physiological function of SPRING by generating and characterizing a line of liver-specific Spring knockout mice. These mice showed significant reductions in levels of active SREBP and SREBP-regulated lipogenic gene expressions. As a result, hepatic de novo lipogenesis and hepatic lipid contents were reduced in liver-specific Spring knockout mice. The work described here is straightforward and the clear cut results in general support the conclusion. The findings described, however, are largely predictable and do not provide any conceptual advance. The lack of novelty makes this work more appropriate for a specialty journal.

We thank the reviewers for their suggestions and constructive comments. Please find below our detailed point-by-point response

Reviewer #1

“In this manuscript, Hendrix and colleagues examine the hepatic function of SPRING using Albumin-Cre liver knockout (LKO) mice. The study is very highly quality and clearly demonstrates a role for SPRING in regulation of hepatic SREBP activity as expected from cell culture experiments and zebrafish data from others. The study provides strong evidence that SPRING plays an important physiological role in supporting SREBP pathway activity [...] Overall, the manuscript is very well written, the data support the conclusions and the experimental methods are clear. Below are a few comments that may strengthen the manuscript.”

We thank the reviewer for recognizing the quality and clarity of our work.

“1. The authors demonstrate that "hepatic SPRING is indispensable for intact SREBP signaling". However, they do not present any experiments to test whether SPRING function is regulated and that this impacts SREBP activity. Thus, the use of the word "regulation" in the title is inappropriate, and the title should be modified to something like "Hepatic SREBP signaling requires SPRING ...”

The reviewer is correct in that we do not investigate the regulation of SPRING itself. In our earlier work we reported that expression of SPRING itself is not sensitive to sterol-dependent regulation (PMID: 32111832). While we do not think that the use of “regulation” in the title grammatically indicates that SPRING is being regulated, we agree with the reviewer’s suggestion to refer to the fact that SPRING is required for hepatic SREBP signaling and have changed the title accordingly.

“2. The authors demonstrate that hepatic lipoprotein secretion is decreased in SPRING LKO animals. Given the strong reduction in PCSK9 mRNA and protein, it is also likely that serum cholesterol is reduced due to increased LDLR activity. The authors should either test this directly in mice or at a minimum present it as a likely contributor to the phenotype in the results or discussion.”

As the reviewer correctly indicates, in Figure 1F we show that expression of *Pcsk9* is reduced. Yet we point out that this is also accompanied by a reduction in hepatic expression of the *Ldlr* (Figure 1F,G). This is expected as both are transcriptional targets of SREBP2. As such, transcriptional and post-transcriptional processes controlling abundance of hepatic LDLR are concomitantly modified. As shown in Figure 2D and Figure 2F the level of the LDLR is unchanged in livers and primary hepatocytes of SPRING^{LKO} mice, respectively. This is in stark contrast to other SREBP targets, which are dramatically lower. This suggests that reduced hepatic expression of the *Ldlr* is counter balanced by attenuated post-transcriptional PCSK9-mediated degradation of the receptor. We have already commented on this scenario on p. 14 of the manuscript, and in the revised manuscript further clarified this point. However, as we cannot formally rule out enhanced hepatic clearance, as the reviewer mentions, we also include a comment on this possibility in the “Results” section.

“3. Is there a SPRING antibody that detects hepatic SPRING? If so, levels of SPRING in LKO mice should be tested.”

We have tested multiple commercially available antibodies against SPRING (Invitrogen #PA5-55459, Merck HPA026905, Novus NBP1-82144). Unfortunately, and to our great frustration none of these detect endogenous SPRING in cells or tissues. As such, we rely on *Spring* mRNA expression as a proxy for its level.

“4. In Fig. 2B, please indicate other proteins with significant expression changes. Depending on the identities, please discuss. Supplemental table 5 is not well presented and should better indicate what changes were observed in the MS experiments. What does the score represent? Why is there no fold change column or p value listed?”

The volcano plot in Figure 2B is shown to highlight that a wide range of SREBP targets are significantly reduced in livers of SPRING^{LKO} mice. The full proteomic dataset is provided in modified Supplementary Table 5 and the raw data files are deposited on <https://figshare.com/s/a9bee752a3cd343c2412> and will be made public. Next to the pathway analysis of the proteomics dataset that is presented in Figure 2C, we have also marked a number of additional significantly changed proteins as requested.

We apologize for the confusion regarding the Supplemental proteomics table. The “score” parameter represents the confidence scoring for individual proteins and is determined based on the false discovery rate, protein occurrence, protein sequence coverage, and protein length; the higher the better. In modified Supplementary Table 5 we included (filterable) columns containing the fold-change and *p* value as requested.

“Minor:

1. Please note the p value for panels in Fig 1D.”

Thank you for pointing out the omission. This is now included.

“2. Figure 1E would be clearer if the X-axis indicated the comparison (LKO/WT).”

This is a good suggestion and the X-axis has been modified accordingly.

“3. Please address why LXR target genes may be elevated in SPRING LKO (Fig. 2C).”

The activation of LXR (z-score>2) in SPRING^{LKO} liver was identified using the Ingenuity pathway analysis (IPA™) software package. This algorithm identified activation of the LXR pathway (-log₁₀(*p*-value)=6.35) due to the upregulation of 9 proteins (AMBP, C3, CLU, HPX, PON1, PON3, SERPINA, TF, VTN) and downregulation of 4 proteins (ECHS1, FASN, ACACA, APOB), representing only 10.6% of the pathway-associated proteins (13/123). Given our expertise in studying the LXR pathway we concur that overall these are not typical and canonical LXR targets/changes, yet this was the outcome of the IPA analysis. We added a brief comment on this in the relevant section and point out that the full set of proteins used for calling altered pathways is also included in new Supplementary Table 6.

“4. Supplemental Tables 4&5 need better titles to explain the data within. Supplemental Table 4 should include the p values for all GO term analyses to confirm that those presented in 2C are the most significant.”

We have modified Supplemental Table 4 (RNAseq) and Supplemental Table 5 (proteomics) so they are clearer and as is also detailed in answer 4 above. Additionally, the raw proteomics and RNAseq datasets have been deposited in public repositories and will be made open upon publication (proteomics: <https://figshare.com/s/a9bee752a3cd343c2412> ; RNAseq: reviewer token: stwdsmeojdadpsd , <https://www.ncbi.nlm.nih.gov/geo/query/acc.cgi?acc=GSE236045>). As requested by the reviewer, we have added new Supplementary Table 6 which lists the pathway-associated genes and *p* values for all the identified pathways listed in Figure 2C.

Reviewer #2

“This is a very strong paper that investigates the role of the newly identified SPRING protein in regulating hepatic lipid metabolism. [...] This review focuses on the genetic analysis provided. In summary, the authors present compelling GWAS data and should further investigate said data. Conversely, the missense and loss-of-function variant analysis is vague and speculative and detracts from the overall story. The authors should stay away from speculating about coding variants of unknown consequences.”

Thank you for the insightful comments on the analysis of the human genetics section. We appreciate the comments and have modified the revised manuscript in their spirit.

“The GWAS data from the GLGC and the UK Biobank convincingly demonstrates that there is a genetic signal at the C12orf49 locus for HDL-C and ApoA1 levels in humans. This is the strongest piece of genetic evidence provided by the authors, and they should focus their attention here. There are clearly both coding and non-coding variants in this GWAS signal, and due to linkage it is hard to assign causality (identification of the causal variant is likely beyond the scope of this paper). The authors should investigate eQTL data to see if any non-coding variants associate with gene expression of C12orf49 or other neighboring genes. Any eQTL evidence would strengthen this argument.”

We agree with the reviewer that the GWAS evidence linking SPRING (C12ORF49) to HDL-C and ApoAI in humans is convincing. As suggested, we evaluated if the 17 SNPs significantly associated with HDL-C plasma levels (shown in Figure 6A and listed below) were associated with changes in *C12ORF49* gene expression (or of any other genes). As shown in Figure I for reviewers, none of the SNPs were associated with hepatic *SPRING/C12ORF49* gene expression. Also, none of these SNPs are reported as eQTL in the GTEx dataset (<https://gtexportal.org> ; Current Release (V8)). We have also included a comment on this in the relevant “Results” section.

List of significant *SPRING* variants ($p < 5.0E-08$) evaluated:

rs75853952; rs11836095; rs118051522; rs2644692; rs10507274; rs339452; rs339451; rs339449; rs339448; rs180304; rs180305; rs339459; rs339458; rs339456; rs339455; rs7976188; rs11068167

“The authors could also complement their AlphaFold analysis with predictive analysis of the consequences of the GWAS missense mutations using predictive tools (SIFT and PolyPhen are two tools but there are others) to see if any of the common variants are predicted to be damaging to SPRING function. This and the approach outlined in the first comment would help strengthen the argument that this GWAS signal is highlighting SPRING as causal in humans.”

Thank you for the good suggestion. Accordingly, we annotated the 17 SNPs significantly associated with HDL-C plasma levels (with wANNOVAR ; Table I for reviewers). Among these SNPs, 12 are located in introns of *C12ORF49*, 3 are intergenic and only one is an exonic missense variant (rs10507274; NM_024738:exon2:c.A164G:p.Q55R). We assessed the functional consequence of this SNP using 10 different prediction scores (including SIFT and PolyPhen that were suggested, Table I for reviewers). Eight algorithms (of 10) predict this SNP to have benign (low, tolerated or neutral) effects on the protein, while LRT and Mutation taster predicted a deleterious (or disease causing) alteration. Experimental analyses, which are beyond the scope of the current study, are needed to assess the functional consequences of SPRING p.Q55R. In the revised manuscript we indicate which algorithms were used in the “Methods” section and included a brief comment on the outcomes of the predictive tools in the “Discussion” section.

“While the GWAS data is compelling (but could use some extra support), the missense variant and rare variant analysis is not compelling and somewhat distracting. There is no evidence that Q55R is a gain-of-function mutation. No functional data is provided that demonstrates this in vitro or in vivo. rs10507274 could easily be in linkage disequilibrium with a different coding variant or a non-coding variant that is causal (this goes back to the eQTL analysis). Speculation about the consequence of this missense variant is unwarranted and should be excluded unless the authors want to do the extra work required to prove that this variant is causal and/or gain of function.”

We appreciate the reviewer viewing the GWAS data as compelling and agree with the comment regarding the analysis of the missense and rare SPRING variants. Addressing these points, and establishing a possible functional consequence for the Q55R variant (rs10507274) would unfortunately require a substantial effort that goes beyond the scope of the current study. Hence, to avoid unwarranted speculation, we opted, as the reviewer suggested to exclude this section from the “Results” section in the revised version and to avoid any reference to this being a gain-of-function variant.

“The “definite loss-of-function” variants are also not well substantiated. Two of the variants are in a different isoform of SPRING. The authors do not discuss the relevance of this isoform and thus the LOF variants in this isoform are of unknown significance. Does the short isoform have the same function? Is it expressed in hepatocytes? Do mice have this isoform, and is it knocked out in your LKO? At minimum, the short isoform appears to not have a transmembrane domain suggesting a very different function. The authors should discuss these questions if they want to include those two LOF variants.”

We agree with the reviewer that the data on the “definite loss-of-function” variants is not sufficiently substantiated and detracts from the main message of the study. As suggested by the reviewer, we decided therefore to remove this section and any reference to loss-of-function variants in the revised version.

“Given that global loss of SPRING in mice is lethal, it would follow that the same is true in humans. Presumably the humans are heterozygous for these LOF variants (the authors should clarify this in the paper), and these authors previously showed that mice with heterozygous global deletion are viable (Loregger 2020). Did those mice have a plasma lipid phenotype? Is there any reason to believe that these humans should have a phenotype if they are heterozygous? Is C12orf49 haploinsufficient?”

This is an interesting point. While it is plausible to expect that global loss of *SPRING* in humans would be incompatible with life as is the case with mice (Loregger et al, 2020 ; PMID: 32111832), one needs to be careful in making this assumption. As elaborated in the “Discussion” section, global loss of S1P in mice is lethal, but humans with residual activity (~1%) are viable yet suffer from severe skeletal dysplasia (references 61-63 in the revised manuscript).

The reviewer is correct that mice with global heterozygous deletion of *Spring* are viable (Loregger et al, 2020 ; PMID: 32111832). At that time our primary goal was to generate homozygous null mice and we have not measured plasma lipids (or for that matter any other parameters) in the heterozygous mice. Since the global knockout mice could not be obtained we terminated the line and shifted towards tissue-specific conditional alleles to facilitate further experiments. Therefore, unfortunately we are unable to experimentally address the reviewer’s comment in mice and hence feel it would be too speculative to comment on *SPRING* haploinsufficiency in humans. We kindly hope the reviewer appreciates that recreating the line and obtaining a colony for this purpose would pose a severe delay.

“Finally, the two LOF variants in the long isoform are near the end of the SPRING protein. Depending on their exact location based relative to exon structure, these variants could escape nonsense mediated decay and be produced as truncated proteins that retain function or are hypomorphic. There’s lots of interesting questions here but they are likely beyond the scope of this paper. Given all the unknowns about these LOF variants, I’d exclude them from the paper because they don’t add to the authors’ conclusions and raise too many further questions.”

We agree with the reviewer’s comment and thank him for the valuable advices. As suggested in the comment and as discussed above, we excluded data on rare variants from the manuscript.

“Line 519-520 should read “Global Lipids Genetics Consortium.””

Thanks for catching this. We corrected the text.

Reviewer #3

“SPRING (C12ORF49) is a recently identified modulator of cholesterol and fatty acid metabolism. Previous studies (largely carried out in cultured cells, references 10-14 of the current paper) by the authors and others have shown that SRPING played an essential role in the activation of SREBPs (Sterol Regulatory Element Binding Protein), key transcriptional activators of genes controlling cholesterol and fatty acid metabolism. In the current paper, the authors continued their in vitro works to study the physiological function of SPRING by generating and characterizing a line of liver-specific Spring knockout mice. These mice showed significant reductions in levels of active SREBP and SREBP-regulated lipogenic gene expressions. As a result, hepatic de novo lipogenesis and hepatic lipid contents were reduced in liver-specific Spring knockout mice. The work described here is straightforward and the clear cut results in general support the conclusion. The findings described, however, are largely predictable and do not provide any conceptual advance. The lack of novelty makes this work more appropriate for a specialty journal.”

Thank you for appreciating our work, which extends the study of *SPRING* function from *in vitro* cell models to whole animal physiology. The reviewer does not provide any specific

comments for us to address, and we respectfully disagree with the comment regarding predictability, as often in this field *in vitro* predictions fail to (fully) translate *in vivo*. As such, our work cements the physiological role of SPRING in the SREBP pathway and in our opinion this outcome is of relevance for a broad community of lipid researchers in a wide range of fields. Accordingly, a multidisciplinary journal, such as *Nature Communications*, is a perfect home for this study.

Figure I. SPRING1 (C12orf49) normalized gene expression according to genotype of 17 SNPs significantly associated with HDL-C plasma levels in human liver (GTEx portal, Release V8).

CHROM	POS_b37	rsID	REF ALT	Func:refGene	Exon:refGene	Func:refGene	AAChange:refGene	SIFT	PolyPhen2_HDIV	PolyPhen2_HVAR	LRT	MutationTaster	MutationAssessor	FATHMM	PROVEAN	MetaSVM	MetaLR
12	117140971	rs75853952	G T	Intergenic	na	na	na	na	na	na	na	na	na	na	na	na	na
12	117143310	rs11836095	C T	Intergenic	na	na	na	na	na	na	na	na	na	na	na	na	na
12	117145306	rs118051522	T C	Intergenic	na	na	na	na	na	na	na	na	na	na	na	na	na
12	117159954	rs2644692	A G	Intronic	na	na	na	na	na	na	na	na	na	na	na	na	na
12	117160976	rs10507274	T C	exonic	nonsynonymous SNV	C12orf49:NM_024738:exon2:c.A164G:p.Q58R	T:tolerated	B:benign	B:benign	D:Deleterious	D:disease_causing	L:low	T:Tolerated	N:Neutral	T:Tolerated	T:Tolerated	T:Tolerated
12	117162096	rs339452	A T	Intronic	na	na	na	na	na	na	na	na	na	na	na	na	na
12	117162293	rs339451	C T	Intronic	na	na	na	na	na	na	na	na	na	na	na	na	na
12	117163293	rs339449	C T	Intronic	na	na	na	na	na	na	na	na	na	na	na	na	na
12	117164146	rs339448	T C	Intronic	na	na	na	na	na	na	na	na	na	na	na	na	na
12	117164388	rs180304	T C	Intronic	na	na	na	na	na	na	na	na	na	na	na	na	na
12	117164747	rs180305	C T	Intronic	na	na	na	na	na	na	na	na	na	na	na	na	na
12	117165373	rs339459	A G	Intronic	na	na	na	na	na	na	na	na	na	na	na	na	na
12	117165833	rs339458	T C	Intronic	na	na	na	na	na	na	na	na	na	na	na	na	na
12	117168067	rs339456	A G	Intronic	na	na	na	na	na	na	na	na	na	na	na	na	na
12	117168476	rs339455	C G	Intronic	na	na	na	na	na	na	na	na	na	na	na	na	na
12	117170368	rs7976188	C G	Intronic	na	na	na	na	na	na	na	na	na	na	na	na	na
12	117175930	rs11068167	C T	upstream	na	na	na	na	na	na	na	na	na	na	na	na	na

Table I. Prediction of functional consequences of SNPs in SPRING/C12ORF49

REVIEWERS' COMMENTS

Reviewer #1 (Remarks to the Author):

The revised version of the manuscript thoroughly addresses my previous comments.

Reviewer #2 (Remarks to the Author):

This is a resubmission of a manuscript exploring the in vivo role of SPRING and its ability to regulate SREBP signaling and hepatic lipid metabolism. As my initial review focused specifically on the genetic data included in the manuscript, so too does this review. Overall the portions of the manuscript that deal with the human genetics of SPRING are drastically improved, with greater focus on the relevant data and far less speculation that distracts from the important data. Overall I find the manuscript to be greatly improved and I commend the Author's on narrowing the focus of this work. Two minor suggestions:

- 1) This is largely stylistic, but I'd move the AlphaFold modeling results up into the "Results" section where I think it complements all of the other discussion there.
- 2) Thank you for including a statement about the eQTLs. One other detail I would add is, do these SNPs have any eQTLs with genes other than SPRING in the liver? I quickly checked a few and it appears the answer is no. I would include this in the short statement about eQTLs, i.e. "no eQTLs with SPRING or any other genes in the region."

Please find below our detailed point-by-point response

Reviewer #1

“The revised version of the manuscript thoroughly addresses my previous comments.”

Thank you for the constructive and positive comments on our study.

Reviewer #2

“This is a resubmission of a manuscript exploring the in vivo role of SPRING and its ability to regulate SREBP signaling and hepatic lipid metabolism. As my initial review focused specifically on the genetic data included in the manuscript, so too does this review. Overall the portions of the manuscript that deal with the human genetics of SPRING are drastically improved, with greater focus on the relevant data and far less speculation that distracts from the important data. Overall I find the manuscript to be greatly improved and I commend the Author's on narrowing the focus of this work.”

Thank you for the constructive and positive comments on our study that resulted in a more focused presentation of the relevant results.

“Two minor suggestions:

1) This is largely stylistic, but I'd move the Alphafold modeling results up into the "Results" section where I think it complements all of the other discussion there.

The supplementary figure containing the Alphafold modeling is indeed referred to in the “Discussion” section. This model is presented as a potential explanation as to why the identified SPRING_{Q55R} variant may influence the interaction between SPRING and S1P. This is speculative, since no structure of this complex has been reported. Given the inherent uncertainty associated with Alphafold models in general, and specifically with respect to SPRING that lacks any structural homologs we want to be cautious. Moreover, moving the model to the “Results” section would also require moving the associated detailed discussion, which we feel is more suited in the “Discussion” section. Hence, we considered and tried the reviewer’s suggestion yet ultimately decided that leaving the model in the “Discussion” is more adequate and better aligned with the primary focus of our study.

2) Thank you for including a statement about the eQTLs. One other detail I would add is, do these SNPs have any eQTLs with genes other than SPRING in the liver? I quickly checked a few and it appears the answer is no. I would include this in the short statement about eQTLs, i.e. "no eQTLs with SPRING or any other genes in the region."

This is a good suggestion and is in line with our earlier answer. We added the suggested comment to our statement on eQTLs which was already included in the manuscript.